# UniFlow: A Unified Pixel Flow Tokenizer for Visual Understanding and Generation

**Zhengrong Yue**[1,2]    **Haiyu Zhang**[3,2]    **Xiangyu Zeng**[5,2]    **Boyu Chen**[4,7]    **Chenting Wang**[1,2]
**Shaobin Zhuang**[1]    **Lu Dong**[6,2]    **Yi Wang**[2]    **Limin Wang**[5,2]    **Yali Wang**[4,2,*]

[1] Shanghai Jiao Tong University    [2] Shanghai AI Laboratory    [3] Beihang University
[4] Shenzhen Key Lab of Computer Vision and Pattern Recognition, Shenzhen Institutes of
Advanced Technology, Chinese Academy of Sciences
[5] Nanjing University    [6] University of Science and Technology of China
[7] School of Artificial Intelligence, University of Chinese Academy of Sciences

## Abstract

Tokenizer is a crucial component for both visual understanding and generation. To advance toward the ultimate goal of universal modeling, recent research has focused on developing a unified tokenizer. However, existing tokenizers face a significant performance trade-off between understanding and generation, stemming from the inherent conflict between high-level semantic abstraction and low-level pixel reconstruction. To tackle this challenge, we propose a generic and unified tokenizer, namely **UniFlow**, by flexibly adapting any visual encoder with a concise reconstruction decoder. Specifically, we introduce *layer-wise adaptive self-distillation* applied to the well-pretrained visual encoders, which enables UniFlow to simultaneously inherit the strong semantic features for visual understanding and flexibly adapt to model fine-grained details for visual generation. Moreover, we propose a lightweight *patch-wise pixel flow decoder*, which efficiently achieves high-fidelity pixel reconstruction by modeling a conditional flow from the noisy state back to the patch-wise pixel domain. By leveraging the semantic features as visual conditions for the decoder, we effectively alleviate the training conflicts between understanding and generation. Furthermore, the patch-wise learning strategy simplifies the data distribution, thereby improving training efficiency. Extensive experiments across 13 challenging benchmarks spanning 7 widely studied visual understanding and generation tasks demonstrate that UniFlow achieves a win–win outcome. For instance, our 7B UniFlow-XL not only surpasses the 14B TokenFlow-XL by 6.05% on average understanding benchmarks, but also achieves a competitive results in both visual reconstruction and generation, surpassing UniTok by 0.15 in rFID and 0.09 in gFID (without guidance), respectively.

## 1 Introduction

The field of computer vision has witnessed a remarkable evolution, with large-scale models achieving significant success in both visual understanding and generation (Chen et al., 2024b; Peebles & Xie, 2023; Rombach et al., 2022; Batifol et al., 2025; Wang et al., 2025b; Chen et al., 2024a; 2025d;b;g;c; Yue et al., 2025). Vision foundation models (VFMs) (Oquab et al., 2023; Radford et al., 2021; He et al., 2022; Tschannen et al., 2025; Yu et al., 2022; Chen et al., 2025a) have emerged as powerful backbones, offering discriminative semantic representations for a wide range of understanding tasks. Meanwhile, generative models (Kingma et al., 2019; Yu et al., 2021; Peebles & Xie, 2023; Rombach et al., 2022; Sun et al., 2024a) have achieved high-fidelity visual synthesis by distribution modeling approaches. To build more generalist models, researchers attempt to integrate understanding and generation within a single framework (Team, 2024; Wu et al., 2025b; Wang et al., 2024c; Xie et al., 2024b; Deng et al., 2025). However, they depend on different tokenizers for understanding and generation, resulting in divergent optimisation objectives that hinder achieving excellent performance in both tasks. Consequently, recent studies have focused on designing unified tokenizers (Wu et al., 2024b; Ma et al., 2025; Zhao et al., 2025b; Qu et al., 2025; Song et al., 2025).

---

*Corresponding author.

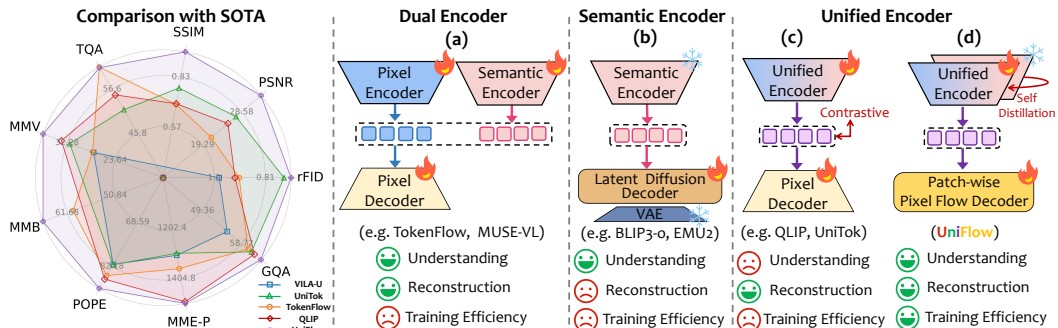

Figure 1: **Comparison of different training paradigms for unified tokenizers.** All multimodal large language models are trained on LLaVA-v1.5 data with Vicuna-7B, except that TokenFlow uses Vicuna-13B. UniFlow simultaneously improves performance and training efficiency.

As shown in Fig. 1 (a), the pioneering methods (Qu et al., 2025; Xie et al., 2024c) employ pixel and semantic encoders to address generation and understanding tasks, respectively. However, this dual-encoder paradigm not only introduces substantial model redundancy but also causes training inefficiency due to the presence of separate embedding spaces. To alleviate this problem, researchers try to design a unified encoder architecture. Some methods (Sun et al., 2024b; Chen et al., 2025f) utilize frozen, well-pretrained vision foundation models as visual encoders and incorporate a latent diffusion decoder for pixel reconstruction, as shown in Fig. 1(b). Although they inherit strong understanding capabilities from vision foundation models, the features extracted by the semantic encoder fail to model fine-grained details, limiting high-fidelity reconstruction. Moreover, the reliance on the pretrained Variational Auto-Encoder (VAE) imposes a ceiling on reconstruction performance. Alternatively, as shown in Fig. 1(c), (Wu et al., 2024b; Ma et al., 2025; Zhao et al., 2025b) initialize visual encoders using pretrained foundation models and finetune them with a pixel decoder by directly mapping semantic tokens to pixel targets for unification. However, this approach may degrade the understanding capacity of visual encoders, as high-level features are simultaneously optimized for low-level reconstruction. Although vision–text contrastive learning is introduced to mitigate this, it is computationally expensive and still struggles to achieve strong understanding capabilities. Hence, this leads to the question: *How can we **efficiently** unify visual representations within a single tokenizer to achieve both **powerful semantic understanding** and **high-fidelity reconstruction**?*

To fill this gap, we propose a generic and unified tokenizer, named **UniFlow**, which efficiently resolves this long-standing trade-off problem via a novel patch-wise pixel flow decoder seamlessly compatible with any semantic encoders. As shown in Fig. 1 (d), UniFlow synergistically integrates these two key components to achieve a optimal balance. Specifically, we leverage a well-pretrained vision foundation model as the encoder. To preserve strong understanding capabilities, we design a layer-wise adaptive self-distillation method that aligns our unified encoder with a frozen encoder, thus preserving hierarchical semantic knowledge, while flexibly complementing its fine-grained representations. Additionally, we propose a novel patch-wise pixel flow decoder to efficiently transform high-level semantic features into the pixel space via Flow Matching (Liu et al., 2022). By modeling a conditional flow directly in the pixel space, we achieve superior reconstruction performance without being constrained by the pre-trained VAE's limitations. The patch-wise learning strategy further reduces the learning burden, thereby improving training efficiency. As a result, UniFlow effectively alleviates the optimization conflict, enabling the encoder to concentrate on discriminative representation learning, while the decoder excels at high-fidelity reconstruction guided by high-level semantic features. Thanks to the well-pretrained encoder and lightweight decoder, UniFlow serves as a general ***unified adaptation paradigm*** that fits any pretrained encoder, whether standalone VFMs or visual backbone of MLLMs, in only 30 ImageNet training epochs.

We conduct extensive experiments on 13 challenging benchmarks across 7 mainstream tasks, including understanding tasks (*i.e.*, visual question answering, image classification, semantic segmentation, depth estimation, object detection) and generation tasks (*i.e.*, image generation, image reconstruction), to demonstrate UniFlow's effectiveness. For example, our 7B UniFlow-XL, trained with 40% less data, surpasses the 14B TokenFlow-XL by 6.05% on overall average understanding benchmarks. Furthermore, UniFlow demonstrates superior performance in visual reconstruction and generation, achieving a new state of the art in reconstruction by outperforming UniTok by 0.15 and SD-VAE by 0.41 in rFID, and competitive results in generation (gFID better than UniTok by 0.09 without guidance). These results demonstrate that UniFlow achieves a win–win outcome, confirming its versatility in both visual understanding and generation.

## 2 RELATED WORK

**Visual Tokenizer for Generative Modeling.** Visual tokenizers are widely used by modern generative models (Rombach et al., 2022; Labs, 2024) to obtain compact latent representations, a process that greatly reduces computational complexity. Some methods improve reconstruction quality via KL constraints (Kingma et al., 2019) or enhancing codebook utilization (Luo et al., 2024; Mentzer et al., 2023; Yang et al., 2021), while yielding suboptimal semantic representations for multimodal understanding. Others attempt to enrich latents with semantic information by aligning features from powerful pre-trained models (Yao et al., 2025; Li et al., 2024c; Chen et al., 2025e). However, their weak alignment fails to preserve the semantic integrity of the original models. Tokenizers based on diffusion or flow matching decoders (Yang et al., 2025b; Shaulov et al., 2025; Wang et al., 2025a) are constrained by a frozen VAE latent space, hindering high-fidelity reconstruction. While these methods preserve local details, they often struggle to capture rich high-level semantic context.

**Unified Tokenizer for Understanding and Generation.** Recent approaches (Wu et al., 2024b; Ma et al., 2025; Wu et al., 2025d) explored unified vision encoders aligning features for both tasks, yet their single-flow architecture rigidly constrains high-level semantic and low-level pixel representations, causing inherent objective conflicts that limit performance. To address this, others used dual encoders or multi-layer representations from a single encoder to handle semantic understanding and pixel reconstruction separately (Qu et al., 2025; Lin et al., 2025), but this introduced inefficient inference and token redundancy. Additionally, emerging models (Sun et al., 2024b; Chen et al., 2025f) aligned pretrained diffusion models with frozen encoders. However, the frozen encoders struggle to capture fine-grained details, which hinders high-fidelity reconstruction under diffusion frameworks. In contrast, UniFlow addresses these limitations via layer-wise self-distillation coupled with a pixel-level flow decoder.

## 3 METHOD

Our **Uni**fied Pixel **Flow** Tokenizer (**UniFlow**) is a novel autoencoder architecture designed to resolve the inherent trade-off between semantic understanding and high-fidelity pixel reconstruction. As illustrated in Fig. 2, UniFlow consists of a unified encoder $\mathcal{E}_{\mathrm{U}}$ and a lightweight flow-based decoder $\mathcal{D}_{\mathrm{flow}}$. The encoder preserves the hierarchical semantic knowledge of a pre-trained encoder via *Layer-wise Adaptive Self-Distillation* (Sec. 3.1). Unlike classical autoencoders, we adopt a lightweight *Patch-wise Pixel Flow Decoder* to reconstruct high-fidelity pixel in a patch-wise manner conditioned on semantic features (Sec. 3.2).

### 3.1 LAYER-WISE ADAPTIVE SELF-DISTILLATION

A robust unified encoder must possess a dual capability: *low-level pixel details* for high-fidelity reconstruction and *high-level representations* for semantic understanding. These competing demands create an inherent conflict for the encoder (Song et al., 2025), making it difficult to fulfill them. For instance, approaches that distill only the final layer (Tang et al., 2025) impose weak constraints, risking semantic degradation and requiring complex multi-stage training. Meanwhile, large-scale contrastive learning methods (Zhao et al., 2025b; Ma et al., 2025) face inherent conflicts between global features and local details, remaining prone to distribution shifts even with high training costs.

To overcome these limitations, we propose a layer-wise adaptive self-distillation method inspired by prior observations Song et al. (2025); Lin et al. (2025) that deeper layers specialize in semantic disambiguation, whereas shallow layers excel at capturing fine-grained details. We posit that distillation should respect this specialization: *deeper layers require stronger preservation for semantic capabilities, while shallow layers need flexibility for fine-grained reconstruction.* Our method follows this principle by dynamically adjusting distillation strength across layers, bridging semantic stability and reconstruction fidelity. In this way, we not only preserves the powerful and hierarchical semantic representations but also allows the encoder to flexibly complement fine-grained details.

Specifically, we use a student encoder $\mathcal{E}_{\mathrm{U}}$ and a frozen teacher encoder $\mathcal{E}_{\mathrm{T}}$ for distillation. For an input image $\mathcal{I} \in \mathbb{R}^{H \times W \times 3}$, both encoders produce feature maps $\mathbf{H}^{(l)} \in \mathbb{R}^{S \times D}$ at each layer $l$. $S$ denotes the number of spatial tokens and $D$ denotes the channel dimension. Our method fuses two key factors to compute the adaptive layer-wise weights $w_l$. First, a *hierarchical prior* $w_l^{\mathrm{base}} = \frac{l}{L}$,

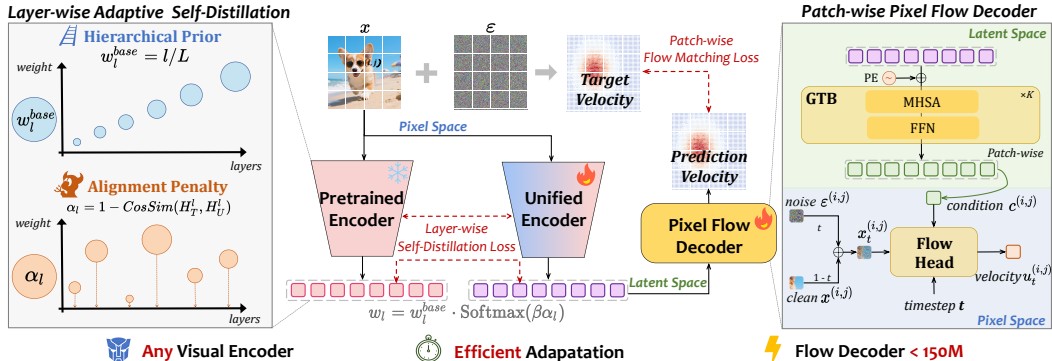

Figure 2: **The framework of UniFlow.** Our UniFlow model is trained end-to-end to endow a powerful VFM with both semantic understanding capabilities and high-fidelity pixel reconstruction.

ensures that deeper layers receive a higher coefficient, where $L$ is the total number of layers. Second, we introduce an *alignment penalty* $\alpha_l$, which measures the average cosine distance between the student tokens and teacher tokens in layer $l$. The adaptive weight $w_l$ is a normalized combination of these two factors, prioritizing poorly aligned layers by assigning a greater weight to those with a higher alignment penalty:

$$w_l = \frac{w_l^{\text{base}} \cdot \exp(\beta \cdot \alpha_l)}{\sum_{k=1}^{L} w_k^{\text{base}} \cdot \exp(\beta \cdot \alpha_k)}, \tag{1}$$

where temperature hyperparameter $\beta$ controls the weight of poorly aligned layers. The self-distillation loss is then the weighted sum of per-layer cosine distances between features:

$$\mathcal{L}_{\text{dist}} = \sum_{l=1}^{L} w_l \cdot \left(1 - \frac{1}{S} \sum_{i,j} \frac{\langle \mathbf{H}_{\text{U}}^{(l,i,j)}, \mathbf{H}_{\text{T}}^{(l,i,j)} \rangle}{\|\mathbf{H}_{\text{U}}^{(l,i,j)}\| \|\mathbf{H}_{\text{T}}^{(l,i,j)}\|}\right), \tag{2}$$

where $(i, j)$ indexes the 2D spatial token location. Finally, the last-layer features of the student encoder $\mathbf{H}_{\text{U}}^{(L)}$ are projected to a compact latent space via a linear projection $\mathbf{z} = \mathcal{P}_{\text{down}}(\mathbf{H}_{\text{U}}^{(L)}) \in \mathbb{R}^{\frac{H}{p} \times \frac{W}{p} \times \hat{d}}$ for subsequent generative modeling.

## 3.2 PATCH-WISE PIXEL FLOW DECODER

Prior diffusion-based tokenizers (Shaulov et al., 2025; Wang et al., 2025a) achieve image reconstruction by modeling a conditional distribution in latent space, but often rely on pretrained VAE decoders. This dependency sets an implicit ceiling on reconstruction fidelity and increases inference-time cost via redundant components. In contrast, our lightweight flow decoder $\mathcal{D}_{\text{flow}}$ directly learns a velocity field in *pixel space*, which not only bypasses the limitations of pretrained VAE decoders, but also simplifies the learning burden and significantly improves training efficiency via patch-wise modeling.

Due to the lack of long-range interactions among individual patches in localized decoding process, patch-wise flow decoder may suffer from "grid artifacts". To address this, we introduce global transformer blocks $\mathcal{GTB}(\cdot)$ of depth $K$. We first lift the latent code $\mathbf{z}$ from the encoder to a higher-dimension space via a linear projection $\mathcal{P}_{\text{up}}(\cdot)$, yielding a set of initial conditional latents. The 2D position embeddings $\mathbf{PE}$ are added to the initial conditional latents being fed into the global transformer blocks,

$$\mathbf{C} = \mathcal{GTB}(\mathcal{P}_{\text{up}}(\mathbf{z}) + \mathbf{PE}). \tag{3}$$

Each global transformer block consists of self-attention and FFN, enabling all tokens to exchange information and perceive a global context. The resulting condition tokens $\mathbf{C} \in \mathbb{R}^{\frac{H}{p} \times \frac{W}{p} \times D}$ are globally coherent, serving as a powerful condition for the flow decoder.

The flow decoder $v_\theta(\mathbf{x}_t, t, \mathbf{c})$, parameterized by $\theta$, is a light-weight MLP network that learns a continuous velocity field in pixel space. This network models the transition between patch-wise data and Gaussian noise, following the principles of Rectified Flow (Liu et al., 2022). The conditional latents $\mathbf{c} \in \mathbb{R}^{p \times p \times D}$ provide a compact representation of the desired visual content. Specifically, given a pixel patch $\mathbf{x}^{(i,j)} \in \mathbb{R}^{p \times p \times 3}$, we define a linear interpolation between the ground truth patch

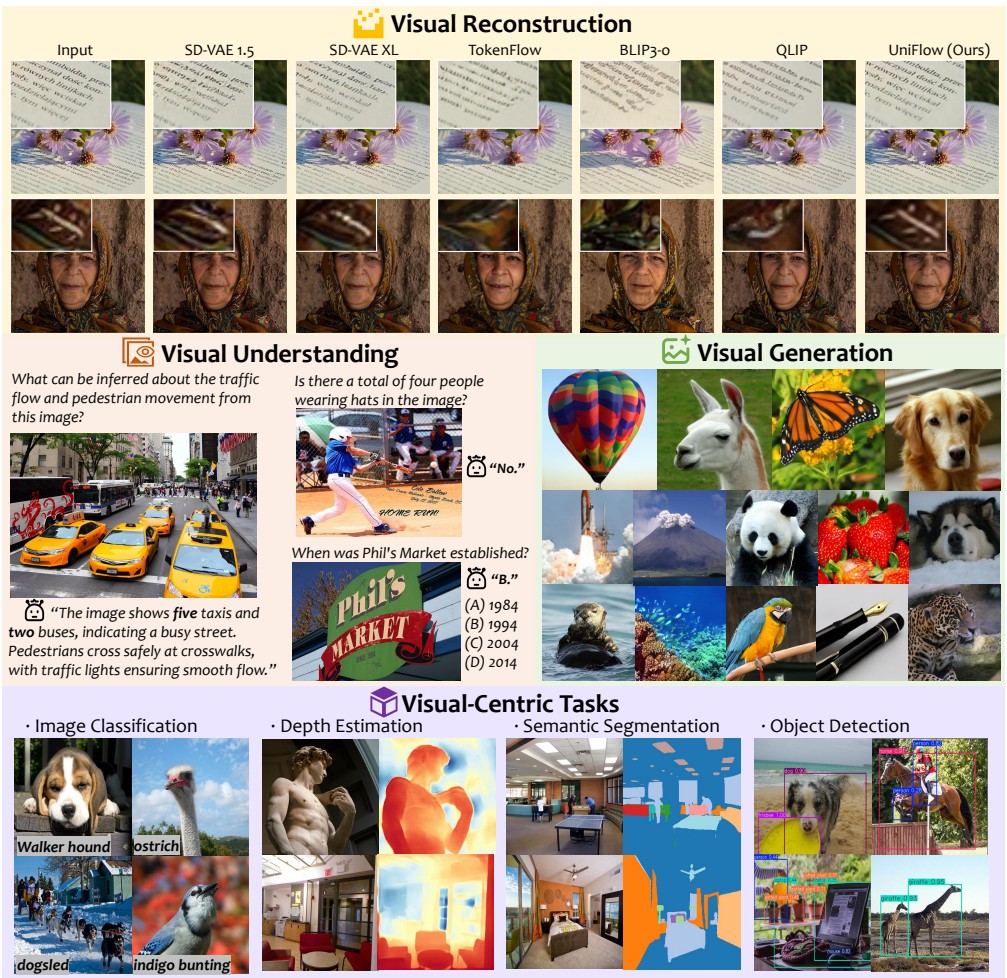

Figure 3: **Various downstream tasks demonstrate UniFlow's robust visual representation.**

$\mathbf{x}^{(i,j)} \sim p_{data}$ and a gaussian noise sample $\epsilon^{(i,j)} \sim \mathcal{N}(0, I)$ at a random timestep $t \sim p_t$:

$$\mathbf{x}_t^{(i,j)} = (1-t)\mathbf{x}^{(i,j)} + t \cdot \epsilon^{(i,j)}, \quad t \in [0, 1] \tag{4}$$

the instantaneous velocity of this trajectory is constant and defined as $\mathbf{u}^{(i,j)} = \epsilon^{(i,j)} - \mathbf{x}^{(i,j)}$. The flow decoder is trained to predict the velocity based on the diffuse time $t$ and the noisy pixel patch $\mathbf{x}_t^{(i,j)}$, along with its corresponding patch-wise conditional latent $\mathbf{c}^{(i,j)}$.

The training objective is to minimize the mean squared error loss to predict the velocity field. The loss applies to each patch, and is formally defined as:

$$\mathcal{L}_{\text{flow}} = \mathbb{E}_{\mathbf{x}^{(i,j)} \sim p_{\text{data}}, \epsilon \sim \mathcal{N}, t \sim p_t} \left\| v_\theta(\mathbf{x}_t^{(i,j)}, t, \mathbf{c}^{(i,j)}) - (\epsilon^{(i,j)} - \mathbf{x}^{(i,j)}) \right\|_2^2. \tag{5}$$

By relying solely on an intuitive flow matching loss, we avoid the complexity of combining multiple losses (*e.g.*, GAN, L1, L2, LPIPS), which leads to more stable training and focus on pixel-leval fidelity. The total training objective of UniFlow is a weighted combination of the Eq. 2 and Eq. 5:

$$\mathcal{L}_{\text{total}} = \lambda_d \mathcal{L}_{\text{dist}} + \lambda_f \mathcal{L}_{\text{flow}}, \tag{6}$$

where $\lambda_d$ and $\lambda_f$ are hyperparameters.

## 4 EXPERIMENTS

### 4.1 IMPLEMENTATION DETAILS

**Datasets.** For the unified tokenizer, we utilize the 1.2M ImageNet-1K (Russakovsky et al., 2014) training set for efficient adaptation training. To enable a fair comparison, we subsequently evaluate

Table 1: **Comparison of reconstruction quality on the $256 \times 256$ ImageNet-1K and MS-COCO 2017 validation sets.** "Ratio" denotes downsampling ratio; "Type" indicates tokenizer traits (VQ usage and decoder type). UniFlow achieves state-of-the-art (SOTA) performance in unified tokenizers while also being competitive with the best generative tokenizers. See Appendix B.1 for data details.

| Method | Type | Training Data | Ratio | ImageNet-1K | | | MS-COCO 2017 | | |
|---|---|---|---|---|---|---|---|---|---|
| | | | | PSNR ↑ | SSIM ↑ | rFID ↓ | PSNR ↑ | SSIM ↑ | rFID ↓ |
| *Generative Only Tokenizer* | | | | | | | | | |
| Cosmos-DI (Agarwal et al., 2025) | Discrete-Pixel | – | 16 | 19.98 | 0.54 | 4.40 | 19.22 | 0.48 | 11.97 |
| LlamaGen (Sun et al., 2024a) | Discrete-Pixel | MS+IN-1K | 16 | 20.65 | 0.54 | 2.47 | 20.28 | 0.55 | 8.40 |
| Open-MAGVIT2 (Luo et al., 2024) | Discrete-Pixel | Mixed100M | 16 | 22.70 | 0.64 | 1.67 | 22.31 | 0.65 | 6.76 |
| BSQ-ViT (Yang et al., 2021) | Discrete-Pixel | 1N-1K | 16 | 28.14 | 0.81 | 0.45 | – | – | – |
| SD-VAE 1.x (Rombach et al., 2022) | Continuous-Pixel | OImg | 8 | 23.54 | 0.68 | 1.22 | 23.21 | 0.69 | 5.94 |
| SD-VAE 2.x (Rombach et al., 2022) | Continuous-Pixel | OImg+LAae | 8 | 23.54 | 0.68 | 1.22 | 26.62 | 0.77 | 4.26 |
| OmniTokenizer (Wang et al., 2024a) | Continuous-Pixel | IN-1K+K600 | 8 | 26.74 | 0.82 | 1.02 | 26.44 | 0.83 | 4.69 |
| SD-VAE XL (Podell et al., 2023) | Continuous-Pixel | OImg+LAae++ | 8 | 27.37 | 0.78 | 0.67 | 27.08 | 0.80 | 3.93 |
| Qwen-Image (Wu et al., 2025a) | Continuous-Pixel | – | 8 | 32.18 | 0.90 | 1.459 | 32.01 | 0.91 | 4.62 |
| SD-VAE 3 (Esser et al., 2024) | Continuous-Pixel | – | 8 | 31.29 | 0.87 | 0.20 | 31.18 | 0.89 | 1.67 |
| Wan2.1 (Wan et al., 2025a) | Continuous-Pixel | – | 8 | 31.34 | 0.89 | 0.95 | 31.19 | 0.90 | 3.45 |
| FLUX-VAE (Labs, 2024) | Continuous-Pixel | – | 8 | **32.74** | **0.92** | **0.18** | 32.32 | **0.93** | **1.35** |
| Cosmos-CI (Agarwal et al., 2025) | Continuous-Pixel | – | 16 | 25.07 | 0.70 | 0.96 | 24.74 | 0.71 | 5.06 |
| VA-VAE (Yao et al., 2025) | Continuous-Pixel | 1N-1K | 16 | 27.96 | 0.79 | 0.28 | 27.50 | 0.81 | 2.71 |
| Wan2.2 (Wan et al., 2025b) | Continuous-Pixel | – | 16 | 31.25 | 0.88 | 0.749 | 31.10 | 0.89 | 3.28 |
| SelfTok (Luo et al., 2024) | Discrete-Diffusion | IN-1K | – | 24.14 | 0.71 | 0.70 | – | – | – |
| FlowMo-Hi (Shaulov et al., 2025) | Discrete-Diffusion | IN-1K | – | 26.93 | 0.79 | 0.56 | – | – | – |
| l-DeTok (Yang et al., 2025a) | Continuous-Diffusion | IN-1K | 16 | – | – | 0.68 | – | – | – |
| *Unified Tokenizer* | | | | | | | | | |
| Show-o (Xie et al., 2024b) | Discrete-Pixel | - | 16 | 21.34 | 0.59 | 3.50 | 20.90 | 0.59 | 9.26 |
| QLIP-B (Zhao et al., 2025b) | Discrete-Pixel | DC-1B | 16 | 23.16 | 0.63 | 3.21 | – | – | – |
| VILA-U (Wu et al., 2024b) | Discrete-Pixel | WL-10B+CY-1B | 16 | – | – | 1.80 | – | – | – |
| Tokenflow (Qu et al., 2025) | Discrete-Pixel | LA+CY | 16 | 21.41 | 0.69 | 1.37 | – | – | – |
| DualViTok (Huang et al., 2025) | Discrete-Pixel | Mixed-63M | 16 | 22.53 | 0.74 | 1.37 | – | – | – |
| DualToken (Song et al., 2025) | Discrete-Pixel | CC12M | 16 | 23.56 | 0.74 | 0.54 | – | – | – |
| MUSE-VL (Xie et al., 2024c) | Discrete-Pixel | IN-1K+CC12M | 16 | 20.14 | 0.646 | 2.26 | – | – | – |
| SemHiTok (Chen et al., 2025i) | Discrete-Pixel | CY-50M | 16 | – | – | 1.16 | – | – | – |
| UniTok (Ma et al., 2025) | Discrete-Pixel | DC-1B | 16 | 27.28 | 0.77 | 0.41 | – | – | – |
| SeTok (Wu et al., 2025d) | Discrete-Pixel | IN-1K+OImg | – | – | – | 2.07 | – | – | – |
| UniLIP (Tang et al., 2025) | Continuous-Pixel | BP-32M | 32 | 22.99 | 0.747 | 0.79 | – | – | – |
| EMU2 (Sun et al., 2024b) | Continuous-Diffusion | LA-CO+LAae | 14 | 13.49 | 0.42 | 3.27 | – | – | – |
| BLIP3-o (Chen et al., 2025f) | Continuous-Diffusion | BP-32M | 16 | 14.71 | 0.58 | 3.18 | – | – | – |
| **UniFlow**(*CLIP*) | Continuous-Diffusion | IN-1K | 14 | 28.66 | 0.91 | 0.67 | 29.61 | 0.92 | 3.69 |
| **UniFlow**(*SigLIP2*) | Continuous-Diffusion | IN-1K | 16 | 29.38 | 0.93 | 0.62 | 26.38 | 0.86 | 3.44 |
| **UniFlow**(*DINOv2*) | Continuous-Diffusion | IN-1K | 14 | 31.01 | 0.94 | 0.54 | 30.66 | 0.94 | 2.81 |
| **UniFlow**(*InternViT*) | Continuous-Diffusion | IN-1K | 14 | **33.23** | **0.96** | **0.26** | **32.48** | **0.95** | **1.88** |

UniFlow's performance on the ImageNet-50K validation set and the MS-COCO 2017 (Lin et al., 2014) validation set. For multimodal understanding, we employ the Pretrain-558K and Instruction-665K datasets as (Liu et al., 2023) for training. For the UniFlow-XL variant, we utilize the approximately 6M subset from LLaVA-OneVision (Li et al., 2024a). We evaluate our models on a comprehensive suite of vision-language benchmarks, including MMVet (Yu et al., 2023), POPE (Li et al., 2023), VQAv2 (Goyal et al., 2017), GQA (Hudson & Manning, 2019), TextVQA (Singh et al., 2019), MMBench (Liu et al., 2024c), and MME (Fu et al., 2023). For visual generation, we train UniFlow on ImageNet-1K. To further verify UniFlow's performance on downstream vision tasks, we perform linear probing experiments for classification, object detection, depth estimation, and semantic segmentation, with models evaluated on ImageNet-1K, MS-COCO 2017, NYU-Depth-v2 (Nathan Silberman & Fergus, 2012), and ADE20K (Zhou et al., 2019).

**Settings.** In our experiments, we employ four variants of the UniFlow Tokenizer, initialized with different semantic teacher models and encoders: DFN-CLIP ViT-L/14-224 (Fang et al., 2023), SigLIP2 ViT-L/16-256 (Tschannen et al., 2025), DINOv2 ViT-L/14-378 (Oquab et al., 2023), and InternViT-300M/14-448 (Chen et al., 2024b) , derived from the pretrained InternVL3-2B-Instruct (Zhu et al., 2025) . The distillation default use $\beta = 2$, while the latent space dimension is set to $\hat{d} = 64$. For lightweight flow decoder, we adopt global transformer blocks of 6 layer followed with an MLP head. All models are trained for 30 epochs with global batch size 512 and fixed learning rate 2e-4. All reported reconstruction performance is based on one-step euler inference. All experiments were conducted on A800 GPUs with PyTorch. More details in the Appendix B.

Table 2: **Evaluation on multimodal understanding benchmarks.** UniFlow-LV indicates training in the LLaVA-v1.5 setting, as marked by †. Our UniFlow-LV achieves SOTA in unified tokenizers. MME is divided by 20 for the Avg.

| Method | VisEnc. | # LLM Params. | Res. | POPE | GQA | TQA | MMV | MMB | MME-S | MME-P | Avg. |
|---|---|---|---|---|---|---|---|---|---|---|---|
| *Understanding Only MLLM* | | | | | | | | | | | |
| InstructBLIP (Dai et al., 2023) | CLIP-G | Vicuna-7B | 224 | – | 49.2 | 50.7 | 26.2 | – | – | – | – |
| MiniGPT-4 (Zhu et al., 2023) | CLIP-G | Vicuna-13B | 224 | – | – | – | – | – | 1158.7 | 866.6 | – |
| InstructBLIP (Dai et al., 2023) | CLIP-G | Vicuna-13B | 224 | 78.9 | 49.5 | 50.7 | 25.6 | 36.0 | – | 1212.8 | – |
| IDEFICS (Laurençon et al., 2024) | CLIP-H | LLAMA-7B | 224 | – | 38.4 | 25.9 | – | 48.2 | – | – | – |
| mPLUG-Owl2 (Ye et al., 2024) | CLIP-L | LLaMA-2-7B | 448 | 86.2 | 56.1 | 58.2 | 36.5 | 64.5 | – | – | – |
| InternVL-Chat (Chen et al., 2024b) | InternViT-6B | Vicuna-7B | 224 | 85.2 | 57.7 | – | – | – | – | 1298.5 | – |
| LLaVA-1.5 (Liu et al., 2023) | CLIP-L | Vicuna-7B | 336 | 85.9 | 62.0 | 46.1 | 31.1 | 64.3 | – | 1510.7 | – |
| Qwen-VL-Chat (Wang et al., 2024b) | CLIP-G | Qwen-7B | 448 | – | 57.5 | – | – | – | 1848.3 | 1487.5 | – |
| LLaVA-OneVision (Li et al., 2024a) | SigLiP-SO400M | Qwen-2-7B | 384 | – | – | 46.1 | 57.5 | 80.8 | 1998.0 | 1580.0 | – |
| *Unified MLLM* | | | | | | | | | | | |
| DreamLLM (Dong et al., 2023) | CLIP-L | Vicuna-7B | 224 | – | – | 41.8 | 22.6 | – | – | – | – |
| LaVIT (Liu et al., 2024b) | CLIP-G | LLaMA-2-7B | 224 | – | 48.0 | – | – | 58.0 | – | – | – |
| Unified-IO 2 (Lu et al., 2023) | VQ-GAN | 6.8B from scratch | 384 | 87.7 | 59.1 | – | 34.3 | 71.5 | 1338.0 | – | – |
| Janus (Wu et al., 2025b) | SigLIP-L | DeepSeek-LLM-1.3B | 384 | 87.0 | 59.1 | – | 34.3 | 69.4 | – | 1338.0 | – |
| LWM (Liu et al., 2024a) | VQ-GAN | LLaMA-2-7B | 256 | 75.2 | 44.8 | 18.8 | 9.6 | – | – | – | – |
| SEED-X (Ge et al., 2024) | Qwen-VL-ViT | LLaMA-2-13B | 448 | 84.2 | 47.9 | – | – | – | – | 1435.7 | – |
| Show-o (Xie et al., 2024b) | MAGVIT-v2 | Phi-1.5-1.3B | 512 | 80.0 | 58.0 | – | – | – | – | 1097.2 | – |
| MetaMorph (Gupta et al., 2022) | SigLIP-SO400M | LLaMA-3.1-8B | 384 | – | – | 60.5 | – | 75.2 | – | – | – |
| Orthus (Kou et al., 2024) | VAE | Chameleon-7B | 256 | 79.6 | 52.8 | – | – | – | – | 1265.8 | – |
| SynerGen-VL (Li et al., 2025) | SBER-MoVQ-GAN | InternLM2-MoE-2.4B | 512 | 85.3 | 59.7 | – | 34.5 | 53.7 | – | 1381.0 | – |
| Liquid (Wu et al., 2024a) | VQ-GAN | Gemma-7B | 512 | 81.1 | 58.4 | 42.4 | – | – | – | 1119.0 | – |
| VILA-U (Lin et al., 2024) | SigLIP-SO400M | LLaMA-2-7B | 384 | 85.8 | 60.8 | 60.8 | 33.5 | – | – | 1401.8 | – |
| Janus-Pro (Chen et al., 2025h) | SigLIP-L | DeepSeek-LLM-7B | 384 | 87.4 | 62.0 | – | 50.0 | 79.2 | – | 1567.1 | – |
| Show-o2 (Xie et al., 2025) | Wan2.1-VAE+ViT-SO400M | Qwen2.5-7B | 432 | – | 63.1 | – | – | 79.3 | – | 1620.5 | – |
| *MLLM with Unified Tokenizer* | | | | | | | | | | | |
| VILA-U † (Wu et al., 2024b) | SigLIP-SO400M | Vicuna-7B | 256 | 81.6 | – | – | – | – | – | 1311.6 | – |
| UniTok † (Ma et al., 2025) | Vitamin-L | Vicuna-7B | 256 | 81.7 | – | – | – | – | – | 1448.0 | – |
| SemHiTok † (Chen et al., 2025i) | SigLIP-L | Vicuna-7B | 256 | 84.2 | 61.0 | – | – | 60.3 | – | 1400.6 | – |
| QLIP † (Zhao et al., 2025b) | CLIP-L | Vicuna-7B | 392 | 86.1 | 61.8 | 55.2 | 33.3 | – | – | 1498.3 | – |
| TokenFlow-B † (Qu et al., 2025) | CLIP-B | Vicuna-13B | 224 | 84.0 | 59.3 | 49.8 | 22.4 | 55.3 | 1660.4 | 1353.6 | 60.21 |
| TokenFlow-L † (Qu et al., 2025) | ViTamin-XL | Vicuna-13B | 256 | 85.0 | 60.3 | 54.1 | 27.7 | 60.3 | 1622.9 | 1365.4 | 62.40 |
| UniTok (Ma et al., 2025) | Vitamin-L | LLaMa-2-7B | 256 | 83.2 | 61.1 | 51.6 | 33.9 | – | – | 1448.0 | – |
| TokLIP (Lin et al., 2025) | VQ-GAN+ViT-SO400M | Qwen2.5-7B | 384 | 84.1 | 59.5 | – | 29.8 | 67.6 | – | 1448.4 | – |
| TokenFlow-XL (Qu et al., 2025) | SigLIP-SO400M | Qwen2.5-14B | 384 | 87.8 | 62.5 | 62.3 | 48.2 | 76.8 | 1922.2 | 1551.1 | 73.04 |
| **UniFlow-LV** † | DFN-CLIP-L | Vicuna-7B | 224 | 86.56 | 61.38 | 53.40 | 30.2 | 63.83 | 1748.0 | 1446.9 | 65.02 |
| **UniFlow-LV** † | SigLIP2-SO400M | Vicuna-7B | 256 | 87.94 | 63.29 | 58.0 | 32.4 | 68.38 | 1823.0 | 1477.9 | 67.87 |
| **UniFlow-LV** † | DINOv2-L | Vicuna-7B | 378 | 88.04 | 59.37 | 45.53 | 25.6 | 51.48 | 1590.5 | 1257.7 | 58.92 |
| **UniFlow-LV** † | InternViT-300M | Vicuna-7B | 448 | 88.97 | 63.35 | 61.85 | 36.6 | 67.10 | 1803.0 | 1505.1 | 69.04 |
| **UniFlow-XL** | InternViT-300M | Qwen2.5-7B | 448 | 89.81 | 65.86 | 81.59 | 54.0 | 83.50 | 2063.0 | 1513.7 | **79.09** |

## 4.2 COMPARISON WITH STATE-OF-THE-ART METHODS

**Visual Reconstruction.** As shown in Tab. 1, our UniFlow method only requires training on ImageNet to achieve state-of-the-art reconstruction performance among unified tokenizers on 256 × 256 ImageNet-1K and MS-COCO 2017 datasets. Notably, UniFlow is also competitive with the best generative-only tokenizers. Specifically, UniFlow(*InternViT*) achieves 0.26 rFID, surpassing UniTok by 0.15 on the ImageNet-1K. These results validate the effectiveness of our pixel-level flow decoder design in preserving fine-grained visual details. Notably, we achieve single-step decoding through our patch-wise decoder design, significantly improving inference speed with high-quality reconstruction. Furthermore, as demonstrated in Tab. 4a, UniFlow exhibits strong reconstruction capabilities at the original resolutions of its respective teacher models.

**Multimodal Understanding.** As shown in Tab. 2, our UniFlow tokenizer consistently demonstrates SOTA performance across a comprehensive suite of multimodal understanding benchmarks. We first evaluate our UniFlow-LV, which consists of four distinct variants trained under the standard LLaVA-v1.5 setting (Liu et al., 2023), each with a different semantic teacher. Using Vicuna-7B as the language backbone, our UniFlow-LV variants consistently outperform prior unified tokenizers such as VILA-U, QLIP, and UniTok across all VQA benchmarks. Notably, the variant using UniFlow(*InternViT*) achieves the highest performance within this group, with a POPE score of 88.97 and an MME-P score of 1505.1, surpassing all others. For the more advanced UniFlow-XL, we train the model under LLaVA-OneVision setting (Li et al., 2024a) but with Qwen2.5-7B (Yang et al., 2024) as the language backbone. UniFlow-XL achieves a new state-of-the-art, which is competitive with or superior to leading approaches that employ larger models and more extensive training data, such as TokenFlow, showcasing the powerful understanding capabilities of our UniFlow tokenizer.

**Text-to-Image Generation.** As shown in Tab. 3, we trained a Multimodal Diffusion Transformer (MMDiT) model to verify UniFlow's generative capability. For the training, we first train

Table 3: Evaluation of text-to-image generation ability on GenEval (Ghosh et al., 2023) and DPG-Bench (Hu et al., 2024) benchmark.

| Methods | Model Size | Type | Res. | GenEval ↑ | DPG-Bench ↑ |
|---|---|---|---|---|---|
| SD v1.5 (Rombach et al., 2022) | 860M | Diffusion | 512 | 0.43 | 63.18 |
| SD v2.1 (Rombach et al., 2022) | 866M | Diffusion | 512 | 0.50 | 64.20 |
| PixArt-$\alpha$ (Chen et al., 2023) | 610M | Diffusion | 512 | 0.48 | 71.11 |
| SANA (Xie et al., 2024a) | 0.6B | Diffusion | 512 | 0.64 | 84.30 |
| SD XL (Podell et al., 2023) | 2.6B | Diffusion | 1024 | 0.55 | 74.65 |
| Show-o (Xie et al., 2024b) | 1.5B | AR+Diffusion | 256 | 0.53 | 67.27 |
| TokenFlow (Qu et al., 2025) | 7B | AR | 256 | 0.55 | 73.38 |
| EMU3-Gen (Wang et al., 2024c) | 8B | AR | 512 | 0.54 | 80.6 |
| **UniFlow** | **0.6B** | **Diffusion** | **256** | **0.65** | **84.76** |

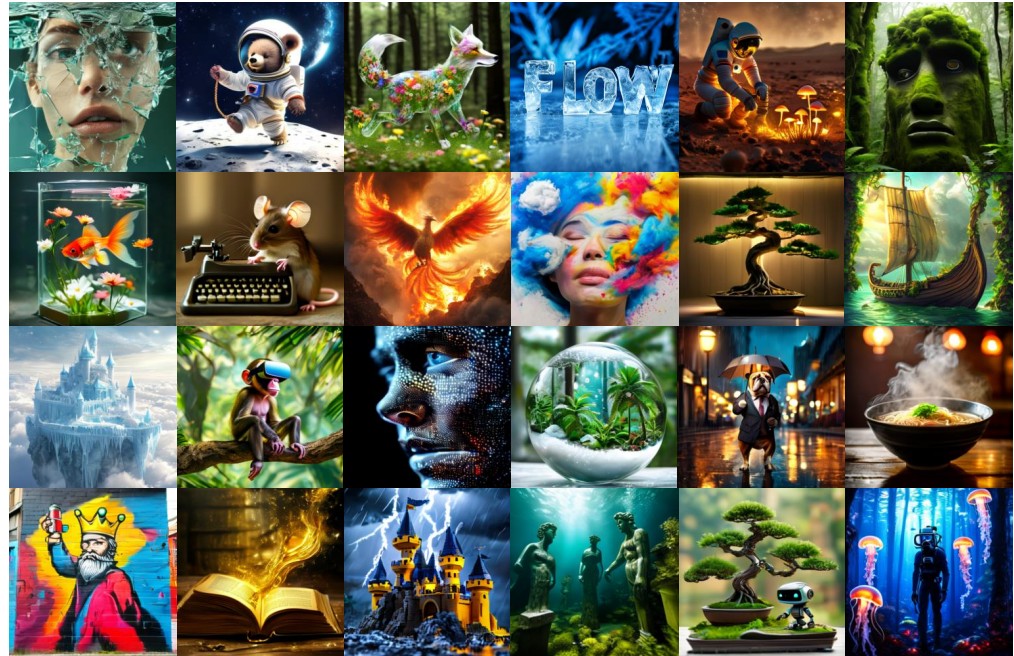

Figure 4: **Text-to-image generation results with UniFlow at 256 × 256.**

a UniFlow(*SigLIP2*)-f16c32 tokenizer with its latent space aligned using DINOv2, and adopted the two-stage training strategy proposed by DC-Gen (He et al., 2025) to initialize the model from SANA-0.6B. This approach allows UniFlow to seamlessly adapt to the T2I task with limited data, outperforming both the strong baseline SANA-0.6B and the larger TokenFlow-7B. Fig. 4 further illustrates the open-domain generation quality achieved by UniFlow. More details can be found in Appendix B.3

**Visual Generation.** To evaluate UniFlow's visual generation, we train MAR-L (Li et al., 2024b) on ImageNet-1K with the UniFlow(*InternViT*) variant as image tokenizer. Images were generated at 448 resolution and resized to 256 for evaluation. As shown in Tab. 4b, UniFlow achieve a lower FID than those using the MAR-VAE without CFG, demonstrating that the incorporation of high-level semantics enhances guidance-free generation performance. Fig. 3 displays the diverse and realistic image generation results. This observation aligns with recent studies (Yu et al., 2025; Ma et al., 2025). More details can be found in Appendix B.4.

**Visual-Centric Tasks.** We conduct comprehensive evaluations of UniFlow's transfer learning capabilities across four visual-centric tasks, with comparative results summarized in Tab. 5. On ImageNet-1K, UniFlow achieves a competitive results for linear probing with 82.6% top-1 accuracy on a ViT-L backbone, surpassing strong baselines like MoCo v3 (Chen et al., 2021) (+4.5%) and MAE (He et al., 2022) (+6.8%) while keeping the encoder frozen. For object detection on COCO,

Table 4: **Image reconstruction (*left*) and class-conditional generation (*right*).**

(a) Image reconstruction performance on ImageNet at pre-training resolutions of VFMs.

| Tokenizer | Res. | PSNR↑ | SSIM↑ | rFID↓ |
|---|---|---|---|---|
| SD-VAE-XL | 224 | 25.72 | 0.75 | 0.90 |
| UniFlow(*CLIP*) | 224 | **29.01** | **0.91** | **0.36** |
| SD-VAE-XL | 256 | 27.37 | 0.78 | 0.67 |
| UniFlow(*SigLIP2*) | 256 | **29.62** | **0.93** | 0.62 |
| SD-VAE-XL | 376 | 26.73 | 0.76 | 0.73 |
| UniFlow(*DINOv2*) | 378 | **30.38** | **0.92** | **0.58** |
| SD-VAE-XL | 448 | 27.49 | 0.7747 | 0.51 |
| UniFlow(*InternViT*) | 448 | **32.48** | **0.95** | **0.28** |

(b) Class-conditional image generation results on ImageNet 256×256. "CFG":classifier-free-guidance.

| Tokenizer | Generator | Type | #Params | gFID↓ *w/o CFG* | IS↑ *w/o CFG* | gFID↓ *w/ CFG* | IS↑ *w/ CFG* |
|---|---|---|---|---|---|---|---|
| VQGAN | LlamaGen | AR | 1.4 B | 14.65 | 86.3 | 2.34 | 253.9 |
| UniTok | LlamaGen | AR | 1.4 B | 2.51 | 216.7 | 2.77 | 227.5 |
| SD-VAE | SiT | LDM | 675 M | 8.61 | 131.7 | 2.06 | 270.3 |
| SD-VAE | REPA | LDM | 675 M | 5.90 | – | 1.42 | 305.7 |
| VQGAN | MAGE | MGM | 230 M | 6.93 | 15.8 | – | – |
| TiTok-L | MaskGIT | MGM | 177 M | 3.15 | 173.0 | 2.77 | 199.8 |
| LFQ | MAGVITv2 | MGM | 307 M | 3.07 | 213.1 | 1.91 | 324.3 |
| MAR-VAE | MAR | MGM | 479 M | 2.60 | 221.4 | 1.78 | 296.0 |
| **UniFlow** | MAR | MGM | 479 M | **2.45** | **228.0** | 1.85 | 290.0 |

Table 5: **Comparison on various visual-centric tasks.**

(a) ImageNet-1K classification linear probing results.

| Methods | Size | $ACC_{lp}$ ↑ |
|---|---|---|
| VFMTok | ViT-L | 69.4 |
| BEiT | ViT-L | 73.5 |
| MAE | ViT-L | 75.8 |
| MAGE | ViT-L | 78.9 |
| MoCo v3 | ViT-H | 78.1 |
| **UniFlow** | ViT-L | **82.6** |

(b) Object detection results on MS-COCO 2017 val.

| Methods | Size | $AP_{ft}$ ↑ |
|---|---|---|
| Supervised | ViT-L | 49.3 |
| MoCo v3 | ViT-L | 54.1 |
| BEiT | ViT-L | 56.2 |
| MAE | ViT-L | 57.5 |
| **UniFlow** | ViT-L | **59.2** |

(c) Monocular depth estimation on NYUv2-Depth.

| Methods | $RMSE_{ft}$ ↓ |
|---|---|
| DORN | 0.509 |
| VNL | 0.416 |
| BTS | 0.392 |
| DPT-Hy | 0.357 |
| **UniFlow** | **0.324** |

(d) Semantic segmentation mIoU on ADE20K.

| Methods | Size | $mIOU_{ft}$ ↑ |
|---|---|---|
| Supervised | ViT-L | 49.9 |
| MoCo v3 | ViT-L | 49.1 |
| BEiT | ViT-L | 53.3 |
| MAE | ViT-L | 53.6 |
| **UniFlow** | ViT-L | **55.4** |

Table 6: **Ablation studies of UniFlow training.** We highlight the default setting.

(a) Distillation strategy

| Distillation strategy | PSNR↑ | rFID↓ | MME-P↑ |
|---|---|---|---|
| Final-layer | 33.41 | 0.25 | 1435.6 |
| Uniform | 30.77 | 0.45 | 1518.2 |
| Progressive ($\beta$=0) | 31.91 | 0.38 | 1495.3 |
| Adaptive ($\beta$=2) | 33.23 | 0.26 | 1513.7 |

(b) Loss balance

| $\lambda_d : \lambda_f$ | MME-P↑ | PSNR↑ | rFID↓ |
|---|---|---|---|
| 1:0 | 1478.6 | – | – |
| $10^2$:1 | 1521.4 | 26.57 | 0.62 |
| 1:1 | 1513.7 | 32.48 | 0.26 |
| 1:$10^2$ | 1453.0 | 32.88 | 0.22 |
| 0:1 | 817.2 | 33.69 | 0.19 |

(c) Decoder design

| Decoder Design | PSNR↑ | SSIM↑ | rFID↓ |
|---|---|---|---|
| $\mathcal{D}_{pixel}$ | 25.12 | 0.7245 | 1.89 |
| $\mathcal{D}_{latent\ flow}$ | 26.48 | 0.7362 | 0.72 |
| $\mathcal{D}_{pixel\ flow}$ | 30.15 | 0.9124 | 0.51 |
| $\mathcal{D}_{pixel\ flow}$ w/ $\mathcal{GTB}$ | 33.23 | 0.9636 | 0.26 |

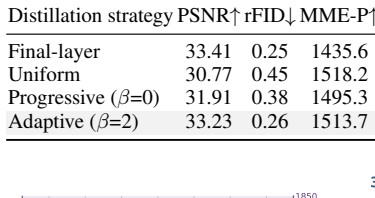

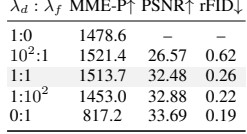

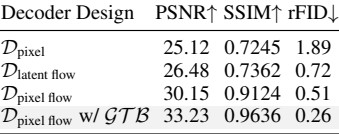

(a) UniFlow(*InternViT*) vs. InternViT

(b) $\beta$ Sensitivity

(c) $\mathcal{GTB}$ Layers

Figure 5: **Ablation studies on training comparison and hyperparameters.**

our method achieves 59.2 AP with a ViT-L backbone, outperforming MAE and BEiT (Bao et al., 2021) by +1.7 and +3.0 points respectively. The flow matching objective's explicit preservation of spatial coherence yields superior fine-grained localization. On depth estimation, UniFlow achieves an RMSE of 0.324 on NYU Depth v2, outperforming DPT-Hybrid (Ranftl et al., 2021) by +10.2%, demonstrating the ability to learn dense features. Finally, for semantic segmentation on ADE20K, we achieve 55.4 mIoU, surpassing MAE and BEiT by +1.8 and +2.1 points respectively. This significant improvement highlights UniFlow's capability to capture both semantic meanings and precise spatial relationships. More experimental details and analysis in Appendix B.5.

## 5 ABLATION STUDY

**Impact of Distillation Strategy.** As shown in Tab. 6a, our ablation study validates the superiority of the adaptive distillation strategy. While final-layer distillation as (Tang et al., 2025) excels at reconstruction and uniform distillation across all layers prioritizes understanding, a progressive baseline ($\beta = 0$) shows significant gains over both by by linearly increasing distillation weights with depth. Ultimately, our layer-wise adaptive distill ($\beta = 2$) achieves the best overall performance by dynamically balancing the two objectives.

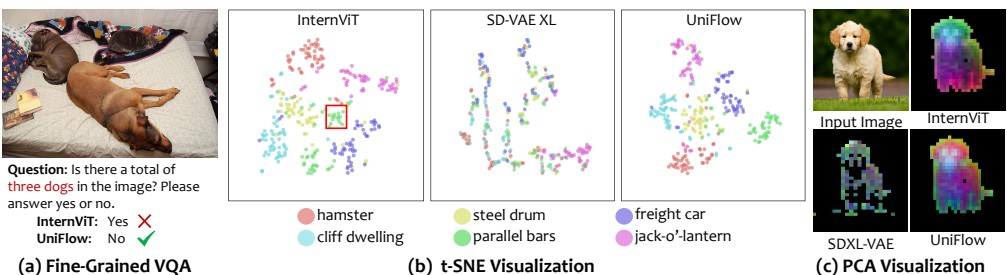

Figure 6: **Qualitative analysis of representations. (a) VQA:** demonstrates UniFlow's superior understanding of detailed concepts. **(b) t-SNE:** UniFlow generates more semantically coherent clusters than InternViT and SD-VAE XL. **(c) PCA:** UniFlow maintains richer spatial information with clearer object contours.

**Effect on Loss Balance.**    Tab. 6b show a clear trade-off between semantic alignment and reconstruction objectives. High $\lambda_d$ prioritizes understanding at the cost of reconstruction, while high $\lambda_f$ yields the opposite. With a balanced loss ($\lambda_d = \lambda_f$), our unified model achieves a near-perfect reconstruction while gaining 35.1 MME-P points over the distillation-only baseline.

**Ablation of Decoder Design.**    As shown in Tab. 6c, we progressively improve the decoder design. A simple pixel loss ($\mathcal{D}_{pixel}$) results in the worst performance. Utilizing a flow model in the latent space ($\mathcal{D}_{latent\ flow}$) offers better results, but it remains constrained by the frozen VAE (Podell et al., 2023). By employing a flow model directly in the pixel space ($\mathcal{D}_{pixel\ flow}$), we achieve a significant performance leap. Finally, the introduction of the Global Transformer Block $\mathcal{GTB}$ eliminates the 'grid effect' observed in the early stages of training and achieves the best overall performance.

**What Does UniFlow Learn?**    To understand what UniFlow learns, we conducted qualitative analyses comparing our UniFlow(*InternViT*) to a semantic encoder (InternViT) and a generative one (SD-VAE XL). As seen in Fig. 6 (a), in a Fine-Grained VQA example, LLaVA-v1.5(*InternViT*) fails to correctly identify the cat in the upper-right corner, mistaking it for a dog. This lack of detail results in misunderstanding, while LLaVA-v1.5(*UniFlow*) captures these details better and gives correct answers. In Fig. 6 (b), t-SNE plots show that while the semantic encoder (InternViT) has clear class clusters and the generative tokenizer (SD-VAE XL) does not, UniFlow's feature space exhibits semantic clustering comparable to InternViT. This demonstrates UniFlow's ability to inherit robust understanding capabilities. Furthermore, in Fig. 6 (c), PCA feature visualizations highlight that UniFlow's features are not only semantically rich but also preserve fine-grained spatial information.

**Comparison with the Teacher Encoder.**    We conducted an ablation study under the standard LLaVA-v1.5 setting, comparing UniFlow(*InternViT*) with its baseline vision encoder, InternViT. As shown in Table 5 (a), by integrating our UniFlow framework, the model not only extended its task range to image reconstruction and generation but also achieved significant performance improvements on core understanding tasks. On benchmarks like GQA and MMB, UniFlow consistently outperforms the original InternViT. We attribute this improvement to our unique layer-wise adaptive distillation and patch-wise pixel flow decoder designs. Specifically, layer-wise adaptive distillation dynamically preserves the powerful hierarchical semantic representations of the pre-trained encoder, while allowing the flow decoder to supplement fine-grained features, thus enhancing the model's visual capabilities without sacrificing understanding performance.

**More Ablation Studies.**    See Appendix D for more ablation analysis about Fig. 5.

## 6    CONCLUSION

In this paper, we proposed **UniFlow**, a unified pixel flow tokenizer designed to address the performance trade-off between visual understanding and visual generation. Our approach integrates a layer-wise adaptive self-distillation strategy for robust semantic preservation and a lightweight patch-wise pixel flow decoder for superior pixel reconstruction. Extensive experiments demonstrate that UniFlow achieves a win-win outcome, proving its effectiveness and versatility in unifying visual representations.

ACKNOWLEDGMENTS

This work was supported by Guangdong Science and Technology Program (Grant No.2024TQ08X365).

ETHICS STATEMENT

Our UniFlow method is developed for advancing representation learnin of generative and understanding, with a focus on supporting creative and beneficial applications. To ensure ethical compliance, our training data is carefully selected from public sources and take deliberate measures to minimize potential biases, fully aligning with universal ethical guidelines. We explicitly emphasize that this framework is not intended for misuse in achieving harmful purposes; downstream users are encouraged to adhere to ethical principles when applying the technology. Additionally, all authors declare no conflicts of interest related to this work.

REPEATABILITY STATEMENT

To ensure full reproducibility, we will publicly release all code and data necessary to replicate our experiments upon paper acceptance. Comprehensive implementation details, including model architecture, hyperparameters, and training methodology, are provided in this paper and its appendix. We are committed to open-sourcing all essential resources to ensure that our findings can be fully verified and built upon by the research community.

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

## A   DIFFERENCE WITH RELATED WORKS

Prior to UniFlow, unified tokenizers for visual understanding and generation were dominated by three mainstream approaches:

(1) **Unified Encoder with a Single-Flow Architecture**. Represented by models such as VILA-U (Lin et al., 2024), QLIP (Zhao et al., 2025b) and UniTok (Ma et al., 2025), these approaches utilize a single encoder to align features for both high-level understanding and low-level reconstruction. These VQ-based solutions typically serve discrete autoregressive (AR) or masked diffusion-based unified models (Team, 2024; Xie et al., 2024b). However, this single-stream design creates an inherent objective conflict that compromises performance. A single network is forced to learn two competing objectives: discarding fine-grained detail for semantic understanding while retaining it for reconstruction.

(2) **Dual-Encoder or Multi-Layer Architectures**. Represented by models like TokenFlow (Qu et al., 2025), SemhiTok (Chen et al., 2025i), DualToken (Song et al., 2025), and Toklip (Lin et al., 2025), these methods address the objective conflict by using separate encoders or different layers of a single encoder to handle understanding and reconstruction tasks independently. While this strategy can successfully separate the two tasks, it introduces significant inefficiencies, including model redundancy, inefficient inference, and token redundancy.

(3) **Encoder-Decoder Alignment with Pre-trained Models**. Represented by models such as Emu2 and BLIP-3o (Sun et al., 2024b; Chen et al., 2025f), these approaches align a pre-trained diffusion model (Rombach et al., 2022; Xie et al., 2024a) with a frozen encoder (Radford et al., 2021; Tschannen et al., 2025). Although they inherit strong understanding capabilities from the encoder, the frozen encoder's features may lack fine-grained details, which hinders high-fidelity reconstruction. Furthermore, a frozen pre-trained VAE also sets an upper limit on reconstruction performance. This combination of factors leads to poor reconstruction capabilities, which directly impairs the fine-grained editing ability of unified models. The inherent diversity of diffusion models, while beneficial for generation, also works against deterministic, high-fidelity reconstruction.

Compared to these three mainstream approaches, UniFlow introduces a unique solution that addresses the limitations of all of them. Our model resolves the fundamental trade-off between understanding and generation by decoupling the two objectives. We design a layer-wise self-distillation strategy that preserves the robust semantic features of a pre-trained encoder for understanding tasks. At the same time, we introduce a separate, lightweight pixel-level flow decoder to achieve high-fidelity reconstruction directly in the pixel space. This design enables our model to achieve state-of-the-art performance in both understanding and reconstruction benchmarks, while also maintaining high training efficiency. As a continuous tokenizer, UniFlow will serve the unified models of AR+Diffusion paradigm, such as BAGLE (Deng et al., 2025), Show-o2 (Xie et al., 2025), etc. Additionally, UniFlow presents an efficient adaptation paradigm that can effectively adapt any visual foundation model into a unified tokenizer, whether it's an independently pre-trained ViT or a vision encoder already integrated with a VLM.

## B   MORE IMPLEMENTATION DETAILS

### B.1   UNIFIED TOKENIZER

**Dataset Abbreviations.**    The specific datasets and their corresponding abbreviations, as used in Table 1, are as follows: YFCC100M (YF), OpenImages (OImg), MS-COCO 2017 (MS), ImageNet-1K (IN-1K), LAION-Aesthetics (LAae), Kinetics-600 (K600), LAION (LA), COYO-700M (CY), DataComp-1B (DC-1B), WebLI (WL), BLIP3o-Pretrain-32M (BP-32M), and LAION-COCO (LA-CO).

**UniFlow Implementation Details.**    The provided tables (7, 8, 9 and 10) detail the training configurations for four UniFlow model variants, each initialized with a different vision-language teacher model: DFN-CLIP ViT-L/14-224 (Fang et al., 2023), SigLIP2 ViT-L/16-256 (Tschannen et al., 2025), DINOv2 ViT-L/14-378 (Oquab et al., 2023), and InternViT-300M/14-448 (Chen et al., 2024b). Since the typical resolution of vision foundation models (VFMs) differs from $256 \times 256$, and to align with their native downsampling ratios ($14\times$ or $16\times$), we train our tokenizer directly on the VFMs'

original resolution. For evaluation, we follow the protocol of (Zheng et al., 2025a) and resize the reconstructed images to $256 \times 256$ to enable consistent quantitative comparison, consistent with the methodology in (Sun et al., 2024a).

Table 7: **UniFlow(*InternViT*) training setting.**

| model | UniFlow(*InternViT*) |
|---|---|
| init weight | InternViT-300M/14 |
| training data | ImageNet-1K |
| image size | [448, 448] |
| data augmentation | random crop, resize |
| downsample | $14 \times 14$ |
| ema | False |
| $\beta$ | 2 |
| encoder depth | 24 |
| $\mathcal{GTB}$ blocks | 6 |
| $D$ (hidden size) | 1024 |
| $\hat{d}$ (latent channel) | 64 |
| flow head depth | 12 |
| flow head width | 1024 |
| flow head patch size | 14 |
| optimizer | AdamW |
| optimizer momentum | $\beta_1, \beta_2 = 0.9, 0.95$ |
| learning rate schedule | consistent |
| learning rate | 2e-4 |
| warmup steps | 0 |
| total epoch | 30 |
| global batchsize | 256 |
| GPU number | 32 A800 |

Table 8: **UniFlow(*CLIP*) training setting.**

| model | UniFlow(*CLIP*) |
|---|---|
| init weight | DFN-CLIP-L/14 |
| training data | ImageNet-1K |
| image size | [224, 224] |
| data augmentation | random crop, resize |
| downsample | $14 \times 14$ |
| ema | False |
| $\beta$ | 2 |
| encoder depth | 24 |
| $\mathcal{GTB}$ blocks | 6 |
| $D$ (hidden size) | 1024 |
| $\hat{d}$ (latent channel) | 64 |
| flow head depth | 12 |
| flow head width | 1024 |
| flow head patch size | 14 |
| optimizer | AdamW |
| optimizer momentum | $\beta_1, \beta_2 = 0.9, 0.95$ |
| learning rate schedule | consistent |
| learning rate | 2e-4 |
| warmup steps | 0 |
| total epoch | 30 |
| global batchsize | 256 |
| GPU number | 32 A800 |

Table 9: **UniFlow(*DINO*) training setting.**

| model | UniFlow(*DINOv2*) |
|---|---|
| init weight | DINOv2-L/14 |
| training data | ImageNet-1K |
| image size | [378, 378] |
| data augmentation | random crop, resize |
| downsample | $14 \times 14$ |
| ema | False |
| $\beta$ | 2 |
| encoder depth | 24 |
| $\mathcal{GTB}$ blocks | 6 |
| $D$ (hidden size) | 1024 |
| $\hat{d}$ (latent channel) | 64 |
| flow head depth | 12 |
| flow head width | 1024 |
| flow head patch size | 14 |
| optimizer | AdamW |
| optimizer momentum | $\beta_1, \beta_2 = 0.9, 0.95$ |
| learning rate schedule | consistent |
| learning rate | 2e-4 |
| warmup steps | 0 |
| total epoch | 30 |
| global batchsize | 256 |
| GPU number | 32 A800 |

Table 10: **UniFlow(*SigLIP*) training setting.**

| model | UniFlow(*SigLIP2*) |
|---|---|
| init weight | SigLIP2-SO400M/16 |
| training data | ImageNet-1K |
| image size | [256, 256] |
| data augmentation | random crop, resize |
| downsample | $16 \times 16$ |
| ema | False |
| $\beta$ | 2 |
| encoder depth | 27 |
| $\mathcal{GTB}$ blocks | 6 |
| $D$ (hidden size) | 1152 |
| $\hat{d}$ (latent channel) | 64 |
| flow head depth | 12 |
| flow head width | 1152 |
| flow head patch size | 16 |
| optimizer | AdamW |
| optimizer momentum | $\beta_1, \beta_2 = 0.9, 0.95$ |
| learning rate schedule | consistent |
| learning rate | 2e-4 |
| warmup steps | 0 |
| total epoch | 30 |
| global batchsize | 256 |
| GPU number | 32 A800 |

## B.2 MULTIMODAL LLMs

Tab.11 details the training configurations for our multimodal LLMs, which are built upon the UniFlow visual tokenizer and its visual features are taken from the second-to-last layer of the UniFlow

Table 11: **UniFlow-LV training setting.**

| model | UniFlow-LV |
|---|---|
| training stage | 2 stages |
| training data | Pretrain(558K) & SFT(665K) |
| vision encoder | all UniFlow variants |
| llm | Vicuna-v1.5-7B |
| optimizer | AdamW |
| optimizer momentum | $\beta_1, \beta_2$=0.9, 0.95 |
| weight decay | 0 |
| warmup ratio | 0.03 |
| max length | 2048 |
| learning rate schedule | cosine |
| learning rate | 1e-3 & 2e-5 |
| total epoch | 1 |
| global batchsize | 256 & 128 |
| GPU number | 8 A800 |

Table 12: **UniFlow-XL training setting.**

| model | UniFlow-XL |
|---|---|
| training stage | 3 stages |
| training data | S1(558K) & S1.5(4M) & S2(2.1M) |
| vision encoder | UniFlow(*InternViT*) |
| llm | Qwen2.5-7B |
| optimizer | AdamW |
| optimizer momentum | $\beta_1, \beta_2$=0.9, 0.95 |
| weight decay | 0 |
| warmup ratio | 0.03 |
| max length | 2048 |
| learning rate schedule | cosine |
| learning rate | 1e-3 & 1e-5 & 2e-6 |
| total epoch | 1 |
| global batchsize | 512 & 512 % 512 |
| GPU number | 32 A800 |

visual tokenizer. The UniFlow-LV model is based on the LLaVA-v1.5 (Liu et al., 2023) framework and uses Vicuna-v1.5-7B (Chiang et al., 2023) as its base LLM. It's trained in two stages with a global batch size of 256 and 128, respectively, on 8 A800 GPUs. For a more powerful variant, the UniFlow-XL model leverages the LLaVA-OneVision (Li et al., 2024a) setting as show in Tab.12, using the Qwen2.5-7B (Yang et al., 2024) LLM and UniFlow(*InternViT*) visual encoder. This model undergoes a more rigorous three-stage training process on a much larger scale, with a consistent global batch size of 512 across all stages on 32 A800 GPUs. Notably, it trains on a 6M subset of the full LLaVA-OneVision dataset.

### B.3 TEXT-TO-IMAGE GENERATION

**Tokenizer.** We first train a new UniFlow (*SigLIP2*)-f16c32 tokenizer based on siglip2-large-patch16-256 on the ImageNet dataset for 40 epochs at 256 resolution. We follow the training paradigm of (Yao et al., 2025), and integrate DINOv2-based latent space alignment to enhance the tokenizer's generative compatibility. As shown in Tab. 13, the achieved reconstruction fidelity confirms its basic ability for high-quality generation.

Table 13: Quantitative Comparison on Reconstruction Quality at $256 \times 256$ Resolution

| Tokenizer | Resolution | rFID ↓ | PSNR ↑ | SSIM ↑ |
|---|---|---|---|---|
| SD-VAE v1.5 | 256×256 | 1.22 | 23.54 | 0.68 |
| DC-AE-f32c32 | 256×256 | 0.69 | 23.85 | 0.66 |
| TokenFlow | 256×256 | 1.37 | 21.41 | 0.69 |
| **UniFlow (*SigLIP2*)** | **256×256** | **0.57** | **28.73** | **0.86** |

**Generator.** For MMDiT training, we utilize 1M high-quality synthetic image-text pairs generated by FLUX-Kera (Labs, 2024) and adopt a two-stage training pipeline as DC-Gen (He et al., 2025): **Stage 1.1** focuses on patch embedder alignment with a learning rate of 1.5e-4 for 15k steps, aiming to bridge the representation gap between the pretrained model and the new latent space; **Stage 1.2** performs joint alignment of the patch embedder and output head with learning rates of 2e-4 and 2.5e-4 respectively for 10k steps, which ensures the consistency of latent reconstruction in the new space; **Stage 2** conducts LoRA fine-tuning with a rank of 256, learning rates of 2.5e-4 for LoRA adapters and 1e-4 for aligned components for 100k steps, designed to adapt the MMDiT to the new latent space while preserving its inherent generative quality.

### B.4 VISUAL GENERATION

We use UniFlow(*InternViT*) as the tokenizer and train the MAR-L (Li et al., 2024b) which are trained with the AdamW optimizer for 400 epochs, using a batch size of 1024 and a learning rate of 1e-5. Diffusion models utilize a linear learning rate warmup followed by a constant schedule, while cross-entropy models are trained with a cosine schedule. Additionally, the exponential moving average (EMA) of model parameters is maintained with a momentum of 0.999. In our specific implementation, The training is performed on 448x448 images, and the model is subsequently resized to 256 for testing. At inference, 256 autoregressive steps are used.

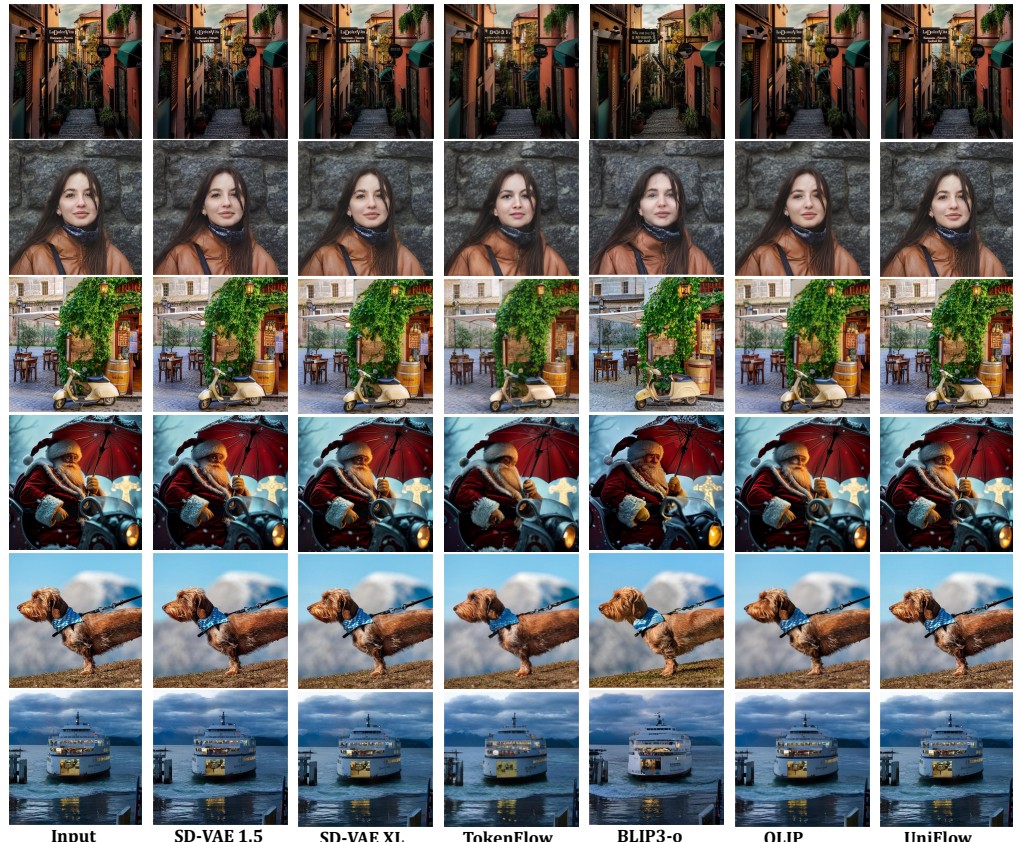

Figure 7: **Visualization of image reconstruction.** All models are inferred on 448 × 448, except for BLIP3-o, TokenFlow, and QLIP, which are inferred on 512, 384, and 392 respectively.

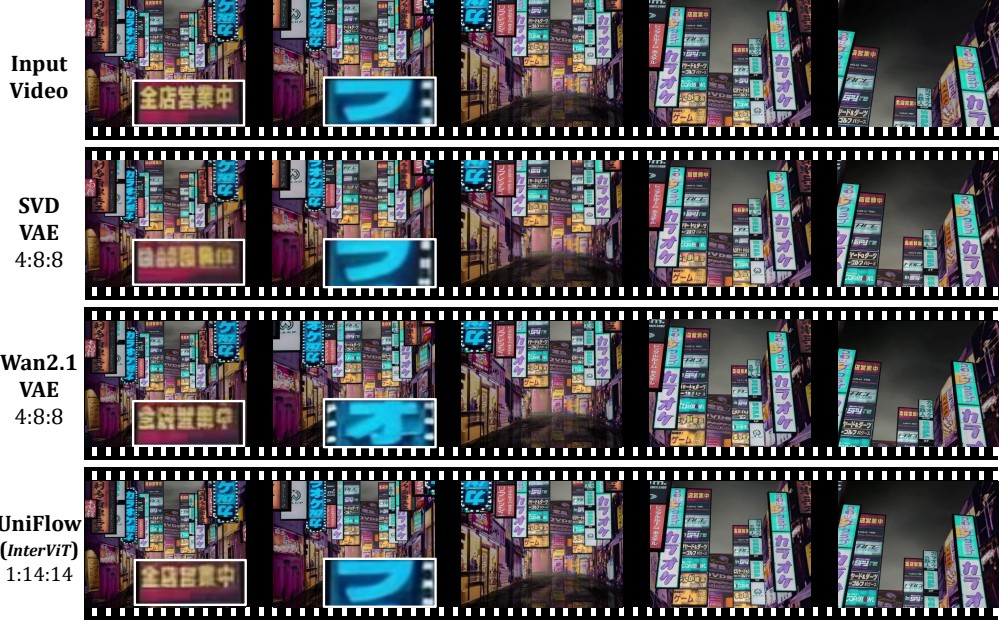

Figure 8: **Visualization of zero shot video reconstruction.** All models are inferred on 448 × 448.

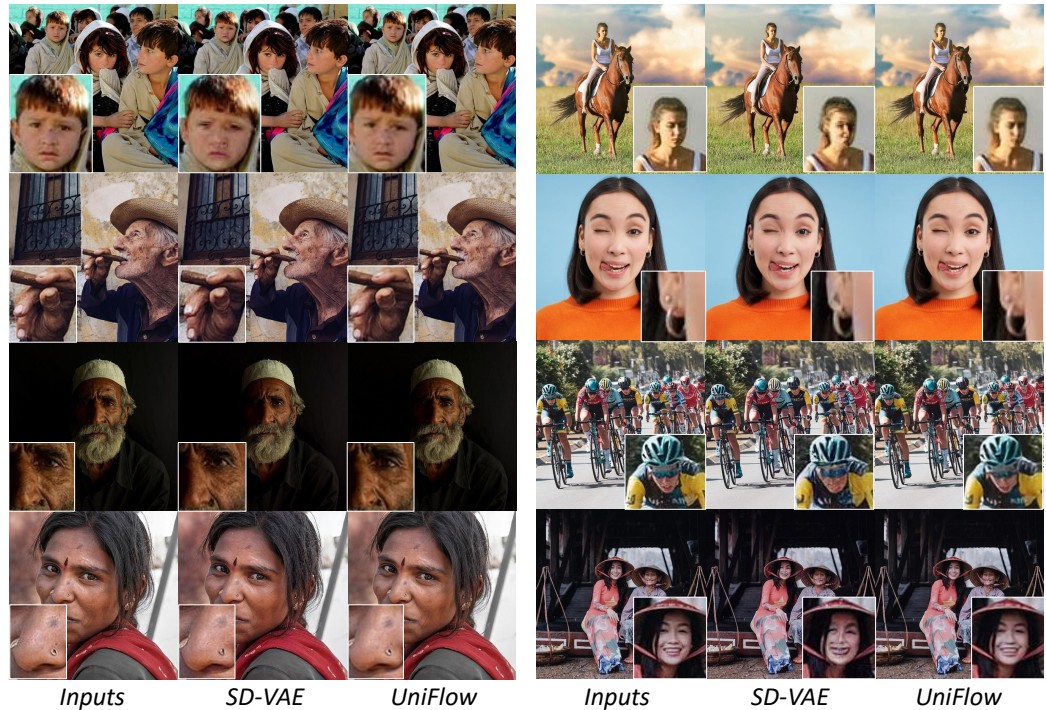

Figure 9: **More out-of-distribution face reconstruction visualization at 448 × 448.**

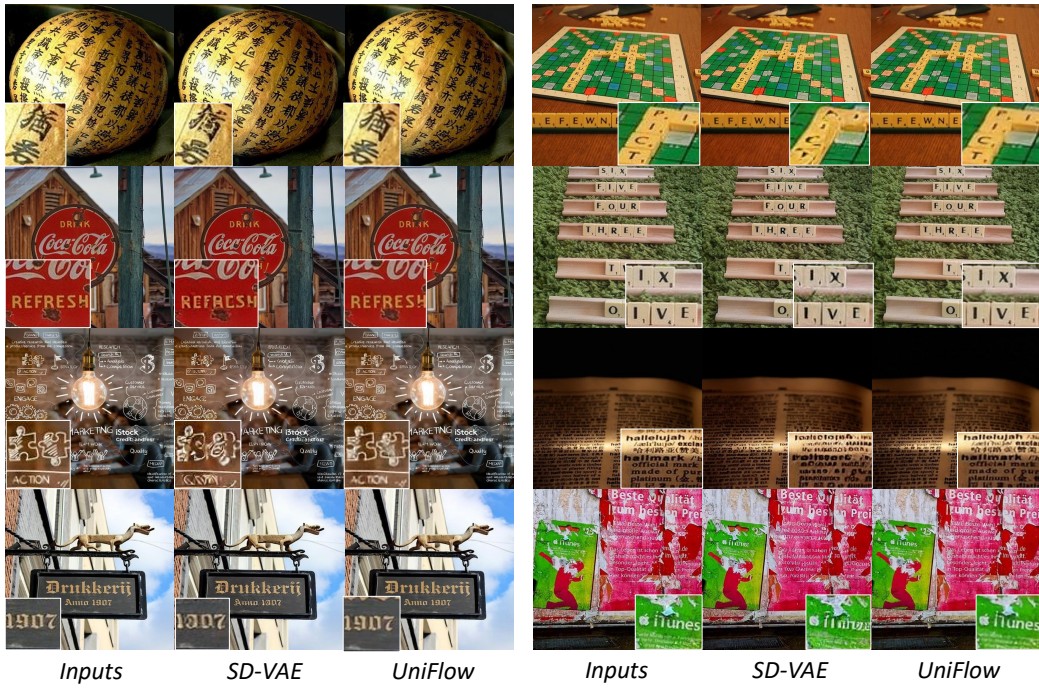

Figure 10: **More out-of-distribution text reconstruction visualization at 448 × 448.**

## B.5 VISUAL-CENTRIC TASKS

**Image Classification.** We follow the protocol of MAE (He et al., 2022) for image classification. Specifically, we evaluate our UniFlow(*InternViT*) model on ImageNet-1K using linear probing. During this process, the UniFlow encoder is frozen, and only the linear classifier is trained. This training is conducted for 100 epochs with a batch size of 128.

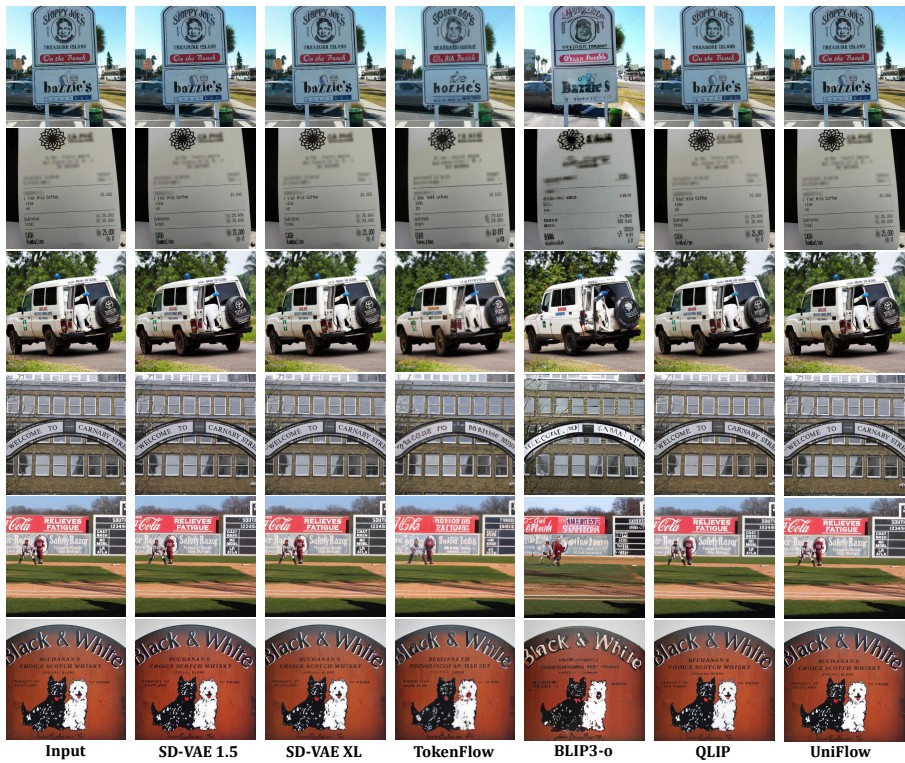

Figure 11: **Qualitative comparison on TokBench text subset.** All models are inferred on 448 × 448, except for BLIP3-o, TokenFlow, and QLIP, which are inferred on 512, 384, and 392 respectively.

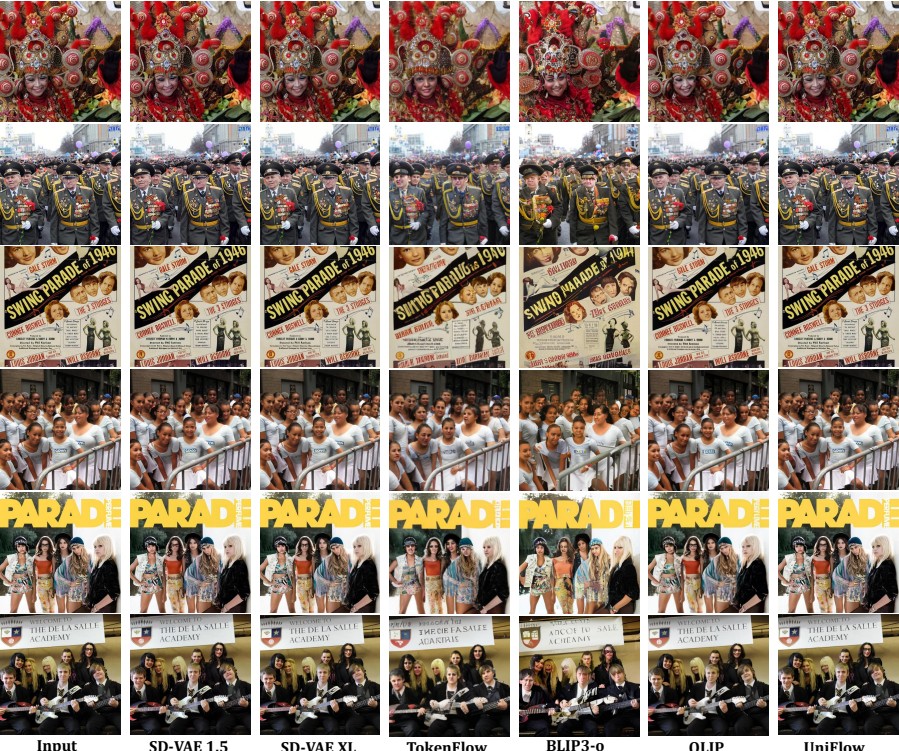

Figure 12: **Qualitative comparison on TokBench face subset.** All models are inferred on 448 × 448, except for BLIP3-o, TokenFlow, and QLIP, which are inferred on 512, 384, and 392 respectively.

**Object Detection.** To validate spatial grounding capabilities, we conduct end-to-end fine-tuning of UniFlow(*InternViT*) on COCO using Mask R-CNN (He et al., 2017) with FPN (Li et al., 2021). We partition the ViT blocks into four distinct subsets and apply convolutional operations to upsample or downsample the intermediate feature maps, thereby generating multi-scale representations. A Feature Pyramid Network (FPN) is subsequently built upon these multi-scale features and trained in an end-to-end fine-tuning manner. The ViT backbone is adaptively modified to be compatible with the FPN structure. We report the standard bounding box Average Precision (AP) metric on MS-COCO 2017 val split. AP denotes the mean Average Precision (mAP) computed at IoU thresholds of [0.5 : 0.05 : 0.95]. All methods use ViT-based backbones with Mask R-CNN architecture.

**Depth Estimation.** To evaluate the quality of UniFlow(*InternViT*) features for monocular depth estimation, we adopt the experimental setup from DPT (Ranftl et al., 2021). Our approach involves a two-stage training process. We first train the model for 60 epochs on the MIX-5 dataset (Ranftl et al., 2022). Subsequently, we fine-tune the model for 20 epochs on the NYU Depth v2 training set. During both stages, we use a constant learning rate of $1 \times 10^{-4}$ and a batch size of 32. The encoder is initialized with pre-trained UniFlow(*InternViT*), while the decoder is randomly initialized. For the model architecture, multi-scale features are extracted from layers [4, 11, 17, 23] of the encoder. During training, input images are resized such that the longer side is 448 pixels, followed by a random square crop of size 448.

**Semantic Segmentation.** For semantic segmentation, we fine-tune our UniFlow(*InternViT*) end-to-end on the ADE20K dataset (Zhou et al., 2019) for 100 epochs. We use a UperNet head (Xiao et al., 2018) with a batch size of 16.

## C  MORE QUALITATIVE RESULTS

### C.1  VISUAL RECONSTRUCTION

**Image Reconstruction.**  Fig. 7 illustrates UniFlow's exceptional static image reconstruction capabilities. Our method consistently generates reconstructions that are remarkably faithful to the original inputs, exhibiting sharp details, accurate textures, and precise color renditions, thereby outperforming other approaches. These results qualitatively validate the overall effectiveness of UniFlow in achieving high-fidelity image synthesis across diverse content. Fig. To further emphasize the performance gap between UniFlow and SD-VAE, more reconstruction visualizations are provided in Fig. 9 and Fig. 10, where UniFlow's superiority is consistently demonstrated.

**Zero-Shot Video Reconstruction.**  In zero-shot video reconstruction, as depicted in Fig. 8, UniFlow demonstrates remarkable proficiency in maintaining temporal consistency and visual quality across video frames without explicit video training. The reconstructed sequences display stable object appearances and fluid motion, showcasing UniFlow's strong generalization and robustness in handling dynamic content. This performance underscores UniFlow's effectiveness in generating coherent visual representations in unseen video domains.

**Qualitative Comparison on TokBench Text Subset.**  For a more rigorous assessment of reconstruction fidelity, Fig. 11 presents UniFlow's qualitative comparison on the challenging TokBench text subset (Wu et al., 2025c). UniFlow achieves superior preservation of intricate text details, where competitor models often struggle with blurriness or distortion. The crispness and legibility of UniFlow's reconstructed characters provide strong qualitative evidence for its ability to capture and reproduce high-frequency information crucial for demanding visual tasks.

**Qualitative Comparison on TokBench Face Subset.**  Fig. 12 offers a qualitative comparison on the TokBench face subset (Wu et al., 2025c), a domain sensitive to perceptual realism. UniFlow consistently delivers reconstructions with superior facial attribute preservation and natural skin textures, surpassing other models which may introduce artifacts or lose subtle expressions. These visual outcomes collectively affirm UniFlow's advanced capacity for high-fidelity generative modeling in perceptually sensitive areas, effectively handling complex and nuanced visual features.

*Is there a yellow plate in the image? Please answer yes or no.*

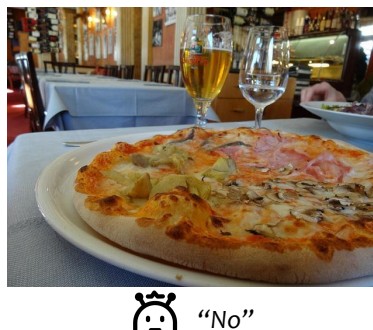

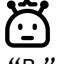 *"No"*

*Is there a remote control on the table? Please answer yes or no.*

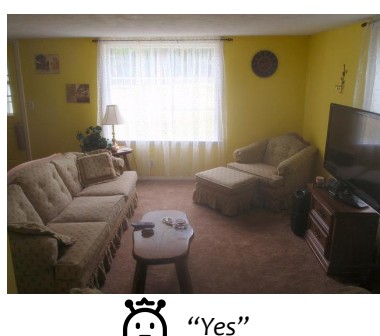

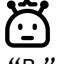 *"Yes"*

*How does Courbet's use of color and brushwork enhance the flowers' beauty?*

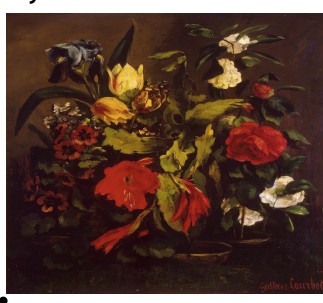

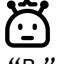 *"Courbet's bold colors and expressive brushstrokes highlight the flowers' vibrancy and fleeting beauty."*

*Which player is in the best position to gain control of the ball first, and what might be their strategy to do so?*

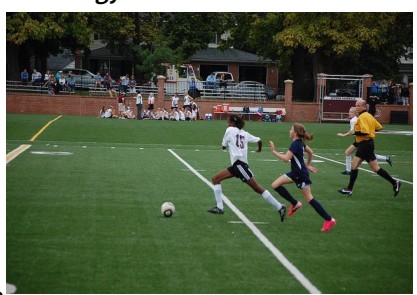

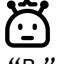 *"Player 15 is poised to control the ball first, leaning forward with an extended foot, likely planning to pass or dribble to advance play."*

*What is the bus number and destination displayed on the bus in the image?*

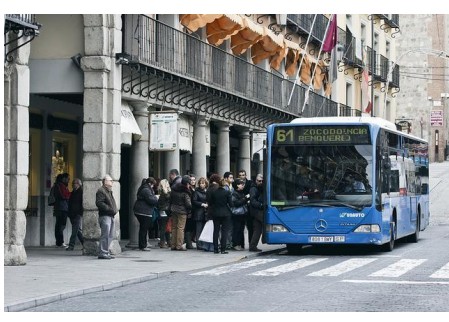

*A) 61 to Zocodonecia Benduero*
*B) 61 to Zocodoncia Benduero*
*C) 51 to Zocodonecia Benduero*
*D) 61 to Zocodoncia Bendiero*

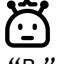 *"B."*

*What time does the watch display in the image?*

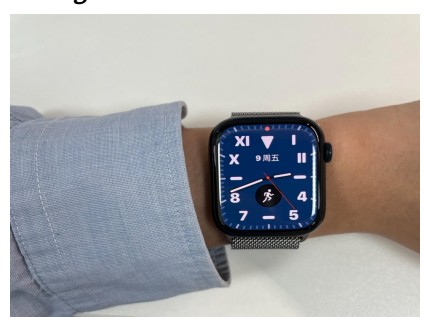

*A) 8:15*
*B) 9:05*
*C) 2:41*
*D) 6:35*

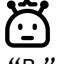 *"C."*

Figure 13: **Visualization of visual question answering.**

## C.2 VISUAL UNDERSTANDING

We provide more multimodal understanding examples in Fig. 13. UniFlow successfully answers the questions accurately. It successfully answers multiple-choice, open-ended, and yes/no questions.

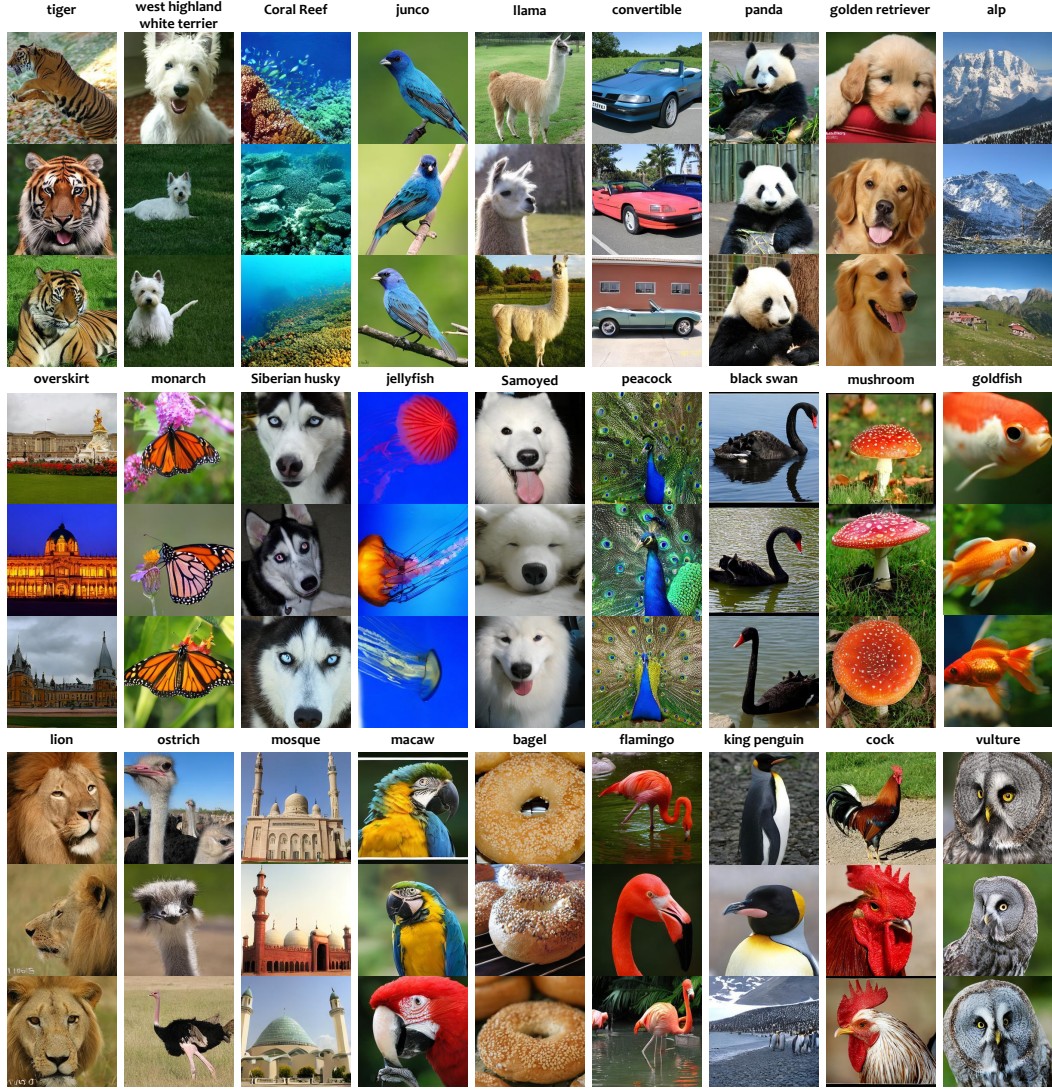

Figure 14: **Visualization of visual generation.**

## C.3 VISUAL GENERATION

We provide more image generation examples in Fig. 14. UniFlow (*MAR-L*) can generate high-quality images given calss.

## C.4 VISUAL-CENTRIC DOWNSTREAM TASKS.

Our analysis confirms the effectiveness of UniFlow on key visual tasks, namely depth estimation, semantic segmentation, and object detection. UniFlow showcases its versatility and robustness in these downstream tasks by demonstrating a strong grasp of both global context and fine-grained details within an image. As shown in Fig. 15, UniFlow accurately infers 3D spatial information from 2D images, producing clear and precise depth maps even for objects and scenes outside its training data. For semantic segmentation, as seen in Fig. 16, the model generates highly precise masks that correctly identify object boundaries and categories. In object detection, illustrated by Fig. 17, UniFlow is capable of accurately localizing and classifying various objects with high confidence. These results collectively confirm UniFlow's position as a highly capable and generalized vision model.

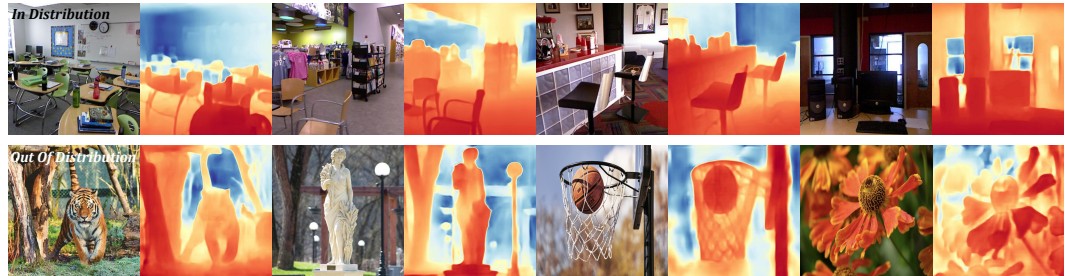

Figure 15: **Visualization of in-distribution depth estimation on NYU-depth-v2 and out-of-distribution depth estimation.**

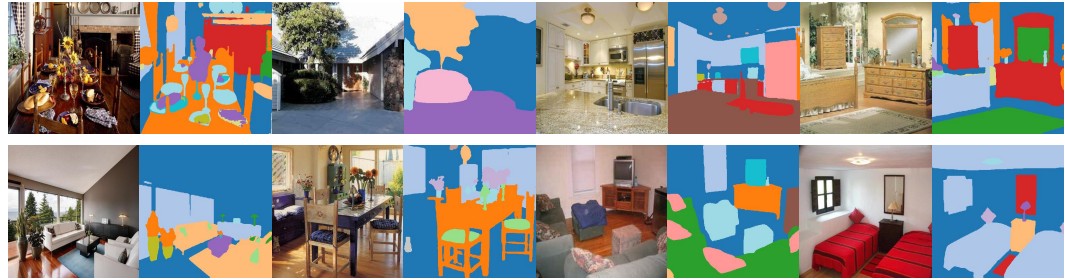

Figure 16: **Visualization of semantic segmentation on ADE20K val.**

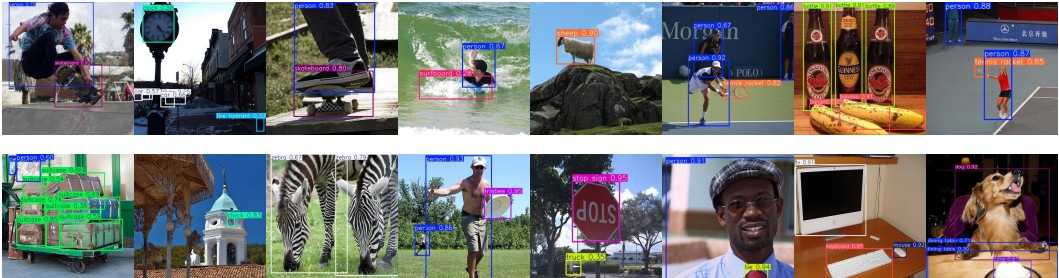

Figure 17: **Visualization of object detection on MS COCO 2017 val.**

## D MORE ABLATION STUDY.

**Training Efficiency Comparison.** To verify UniFlow's training efficiency, we compare it with TokenFlow, BLIP3-o, and UniTok across model size, training data, steps, batch size, and rFID (lower is better reconstruction), as shown in Tab. 14. **Note that BLIP3-o (SigLIP-SANA) is the model released on the BLIP3-o GitHub repository, not the one used in the original paper.** UniFlow uses a compact architecture (InternViT-300M encoder + 145.8M decoder), far less training data (1.2M vs. TokenFlow's 6.6M/BLIP3-o's 32M/UniTok's 1.28B), and only 70k training steps (vs. TokenFlow's 500k/BLIP3-o's 114k/UniTok's 80k), benefiting from its patch-wise decoder that simplifies data distribution. With a 512 global batch size, UniFlow still achieves the best rFID (0.28), outperforming TokenFlow (0.63), BLIP3-o (3.09), and UniTok (0.38). This confirms UniFlow balances efficiency and reconstruction via layer-wise self-distillation and a lightweight decoder.

| Method | Encoder (Backbone) | Decoder Size | Training Data | Training Steps | Global Batch Size | rFID $\downarrow$ |
|---|---|---|---|---|---|---|
| TokenFlow | SigLIP-SO400M | 258.6M | 6.6M | 500k | 256 | 0.63 |
| BLIP3-o | SigLIP2-SO400M | 1771.6M | 32M | 114k | 8192 | 3.09 |
| UniTok | Vitamin-L | 352.4M | 1.28B | 80k | 16k | 0.38 |
| UniFlow | InternViT-300M | 145.8M | 1.2M | 70k | 512 | 0.28 |

Table 14: Comparison of Training Efficiency Across Different Unified Tokenizer Paradigms. The table presents rFID scores, with results for each model measured at its respective training resolution.

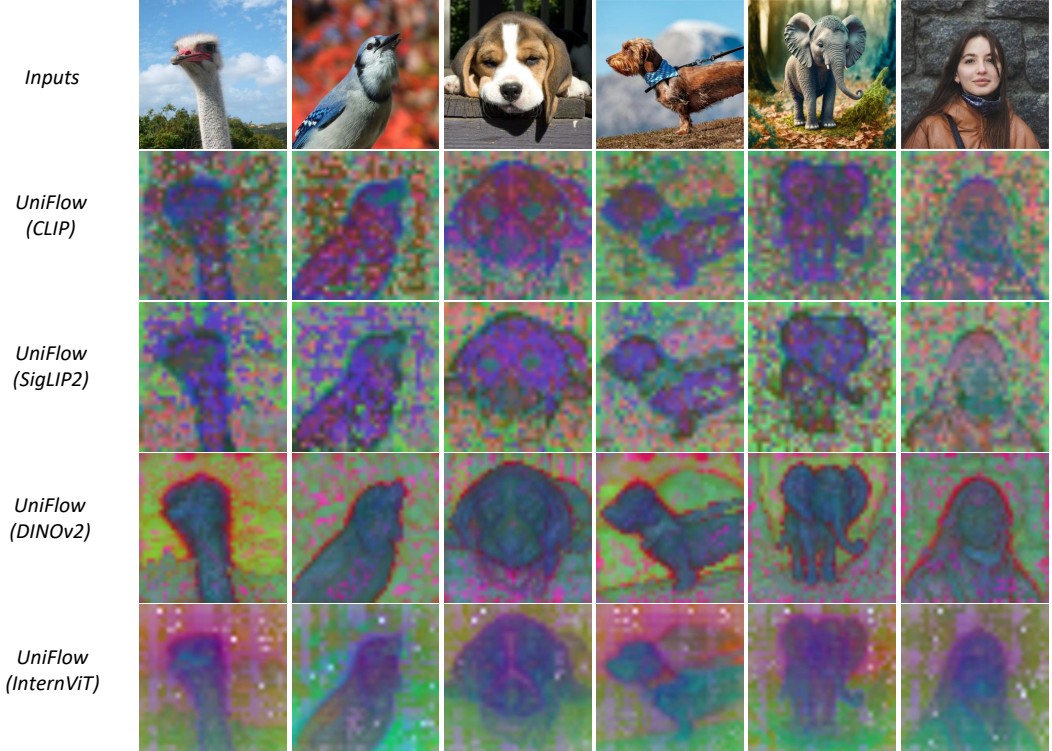

Figure 18: **PCA visualization under different pretrained representations.** The representations of DINO exhibit the most regular distribution, which implies strong generative-friendly properties.

Table 15: **Different pretrained representations (*left*) and Latent Space optimization (*right*).**

| Model | rFID ↓ | gFID ↓ |
|---|---|---|
| SD-VAE | 0.61 | 7.13 |
| DFN-CLIP | 0.67 | 8.35 |
| SigLIP2 | 0.62 | 7.91 |
| DINOv2 | 0.54 | 5.16 |
| InternViT | 0.26 | 7.38 |

| Method | Encoder | Align. | Res. | rFID ↓ | gFID ↓ |
|---|---|---|---|---|---|
| SD-VAE | VAE | None | 256 | 0.61 | 7.13 |
| UniFlow | SigLIP2-SO | None | 256 | 0.62 | 7.91 |
| VA-VAE | VAE | DINOv2 | 256 | 0.28 | 5.14 |
| UniFlow | SigLIP2-SO | DINOv2 | 256 | 0.63 | **5.02** |

**Impact of Pretrained Representations on Visual Generation.** The generative performance of different pretrained encoders and the role of latent alignment are presented in Tab. 15. A core observation is that the quality of generative models is largely determined by the intrinsic properties (structure and distribution) of the latent space inherited from the pretrained encoder, rather than the adaptation framework itself. This is further validated by our ablation study on ImageNet, where models are trained for 64 epochs with LightningDiT-XL. Consistent with findings from recent works such as VAVAE(Yao et al., 2025), RAE (Zheng et al., 2025b) and SVG (Shi et al., 2025), DINO-style representations exhibit stronger generative-friendly properties compared to text-aligned counterparts (e.g., SigLIP, CLIP). Specifically, DINOv2's latent space enables significantly higher fidelity generation with a gFID of 5.16, while the text-aligned InternViT yields a relatively higher gFID of 7.38 (left subtable of Tab. 15). This explains why the initial UniFlow (*InternViT*) does not show a significant advantage: the limitation stems from InternViT's inherent representation characteristics, not UniFlow's adaptation mechanism. UniFlow remains a general framework whose effectiveness is contingent on the teacher encoder's properties. To further explore the potential of the UniFlow framework, we conducted additional experiments at 256×256 resolution. The results (right subtable of Tab. 15) confirm that UniFlow exhibits even stronger generative capabilities when integrated with latent alignment, even for text-aligned encoders. For example, UniFlow (SigLIP2) with DINOv2-based latent alignment achieves a state-of-the-art gFID of 5.02, outperforming both the vanilla UniFlow (SigLIP2) (gFID=7.91) and VA-VAE (gFID=5.14). This demonstrates that while pretrained representations lay the foundation for generative performance, UniFlow with latent

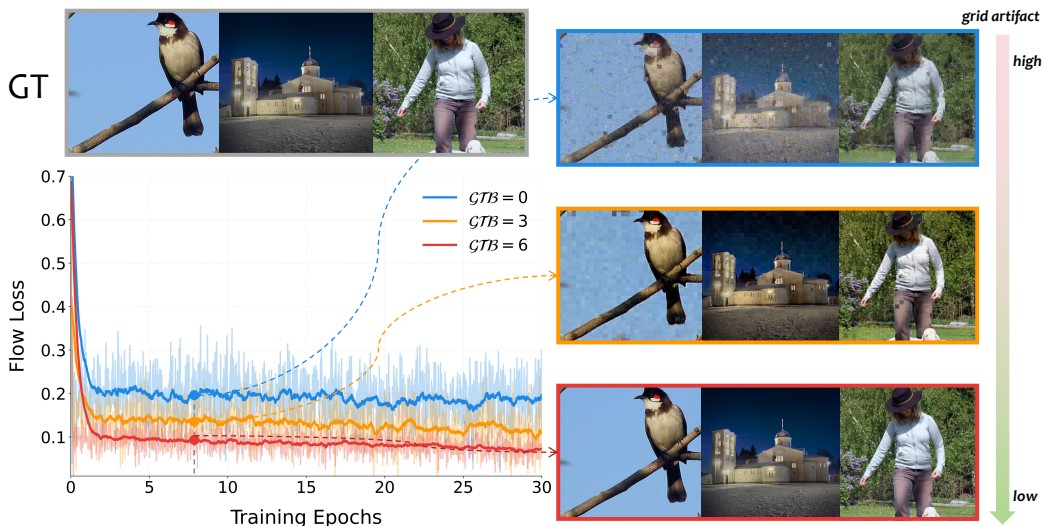

Figure 19: **Visualization of Global Transformer Block (GTB) Impact on Flow Loss and Reconstruction Quality.** The figure shows flow loss curves (left) and corresponding reconstructed images (right) for models with 0, 3, and 6 GTB layers during training. As GTB layers increase, flow loss converges faster and to a lower value, with reconstructed images exhibiting reduced grid artifacts and higher visual fidelity.

alignment can effectively enhance the generative quality of even less generative-friendly (text-aligned) encoders.

**Sensitivity of Temperature Parameter $\beta$.** Fig. 5 (b) presents a detailed sensitivity analysis of the temperature parameter $\beta$ in our adaptive distillation strategy. The results reveal a clear and intuitive trend. A smaller $\beta$ (e.g., 0.5) results in suboptimal performance across all metrics, with a PSNR of 31.99 and MME-P of 1496.2, as it insufficiently penalizes misaligned layers. As $\beta$ increases, the model progressively improves by dynamically emphasizing harder-to-align layers, reaching a robust performance plateau between $\beta = 2$ and $\beta = 3$. Our default setting of $\beta = 2$ achieves an optimal balance, yielding the highest MME-P (1505.1) and a high PSNR (33.23). While a slightly higher PSNR (33.24) is observed at $\beta = 3$, the performance remains stable. When $\beta$ becomes excessively large (e.g., 5), the model over-focuses on correcting misalignments, leading to training instability and performance degradation in both understanding and reconstruction tasks, with PSNR dropping to 32.88 and MME-P to 1489.4. This analysis confirms the necessity of $\beta$ and validates our choice of $\beta = 2$ as a well-balanced and stable configuration.

**Effect of Global Transformer Block.** The Global Transformer Block plays a crucial role in enhancing global consistency and accelerating convergence. As illustrated in Fig. 19, models without sufficient GTB utilization (e.g., 0 layers) suffer from severe grid artifacts in reconstructed images and demonstrate slower convergence during the early training stages, as evidenced by the higher and more volatile flow loss curve. Increasing the number of GTB layers progressively improves both the convergence dynamics and reconstruction quality: the flow loss converges faster to a lower steady - state value, and reconstructed images exhibit reduced grid artifacts and higher visual fidelity. As shown in Fig. 5 (c), the final performance also confirms the block's necessity. Increasing the number of GTB layers progressively improves all metrics, as the model better captures long-range dependencies across patches. Our default setting of 6 GTB layers achieves an optimal balance, yielding a PSNR of 33.23 and an rFID of 0.26. While the performance continues to slightly improve at 9 layers (PSNR of 33.31 and rFID of 0.25), the gains are marginal. This analysis confirms the necessity of the GTB for high-quality reconstruction and validates our choice of 6 layers as the optimal and stable configuration.

**Impact of $\beta$ on Layer-wise Weight Distribution.** Fig. 20 presents a simulated visualization of our layer-wise distillation strategy, designed to illustrate the critical role of the temperature parameter $\beta$ in balancing high-level semantics and low-level details. The x-axis represents the encoder layer

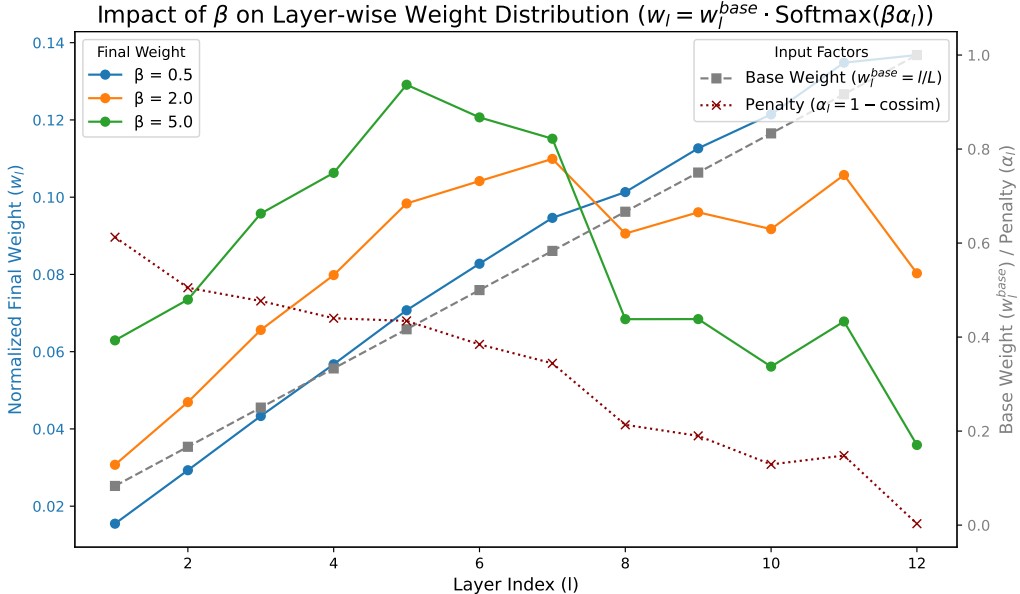

Figure 20: **Simulated Impact of $\beta$ in Layer-wise Distillation.** The figure illustrates how the temperature parameter $\beta$ modulates final distillation weights ($w_l$, left y-axis) across encoder layers ($l$), derived from base weights ($w_l^{\text{base}}$, favoring high-level semantics) and penalty terms ($\alpha_l = 1 - \text{CoSim}$, emphasizing low-level misalignment, right y-axis). Key findings: $\beta = 0.5$ under-amplifies penalties (flat curve, poor low-level utilization); $\beta = 5.0$ over-amplifies penalties (steep curve, instability); $\beta = 2.0$ optimally balances both, enabling effective trade-off between semantics and details.

index ($l$), while the right y-axis shows two key input factors: the linearly increasing base weight ($w_l^{base}$), which intrinsically favors high-level semantics, and the penalty term ($\alpha_l = 1 - \text{CoSim}$), which is highest in low-level layers due to their greater misalignment with the final representation. The left y-axis shows the final distillation weight ($w_l$), a product of these two factors as modulated by $\beta$. As shown, a small $\beta$ (e.g., 0.5) results in a flat weight curve, where the Softmax penalty has little effect. In this case, the final weight distribution closely follows the base weight, failing to adequately utilize fine-grained low-level features for reconstruction. Conversely, an excessively large $\beta$ (e.g., 5.0) leads to a very steep curve, where the penalty term is overly amplified, causing the model to heavily prioritize the most misaligned bottom layers. This over-correction can lead to training instability and compromise the model's semantic understanding. Our analysis confirms that an optimal $\beta = 2.0$ provides the ideal balance, yielding a well-shaped curve that allocates sufficient weight to low-level features to correct deviations and supplement details, while also preserving the crucial high-level semantic information. This demonstrates that $\beta = 2.0$ is the most stable and effective setting for achieving a trade-off between semantic preservation and detail supplementation.

**Ablation on Sampling Steps.** We evaluate reconstruction quality on MS-COCO-val images by sweeping Euler steps from 1 to 100. As plotted in Fig. 21, reconstruction shows a minor downward trend while latency grows nearly linearly with the number of steps, indicating that additional iterative steps do not bring meaningful gains to visual fidelity but instead introduce unnecessary computational overhead. This interesting one-step-optimal phenomenon aligns with prior observations from (Zhao et al., 2025a; Liu et al., 2022): rectified-flow or diffusion-based models rely on multi-step iteration to approximate complex target distributions, but their performance quickly saturates once the conditioning signal is sufficiently determined. For our tokenizer with Flow Matching decoder, the latent vector $\mathbf{z}$ serves as a strong conditional prior—after sufficient tokenizer training, $\mathbf{z}$ has encoded nearly all structural and textural information of the target image, which constrains the conditional distribution $p(\mathbf{x} \mid \mathbf{z})$ to a highly concentrated, near-singleton distribution. In this scenario, the Flow Matching's velocity field can directly model the linear mapping from noise to the target image (as highlighted in Rectified Flow), making one-step sampling sufficient to reach the optimal reconstruction. Additional steps not only fail to refine the distribution but may accumulate

numerical errors from iterative discretization (e.g., Euler method's first-order approximation bias) and cause minor degradation in details like edge sharpness or texture consistency. Consequently, UniFlow retains both minimal latency and maximal reconstruction performance, offering a practical advantage over iterative alternatives in real-world deployment.

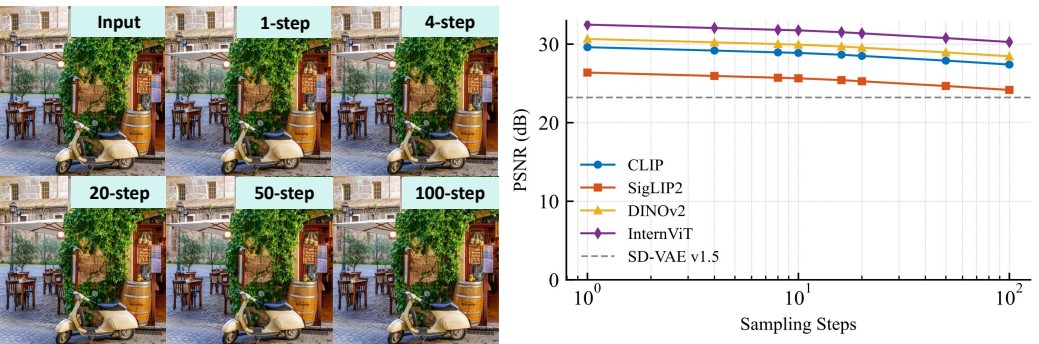

Figure 21: **Impact of sampling steps on reconstruction.**

# E    LIMITATIONS AND FUTURE DIRECTIONS

While UniFlow introduces a new and efficient paradigm for unified visual tokenization, we acknowledge several limitations that also present promising avenues for future research.

First of all, as an academic-driven research model, UniFlow has been primarily validated on controlled, academic benchmarks such as ImageNet due to computational resource constraints. While our method demonstrates strong performance on these standard datasets, there may still be a minor gap in visual quality compared to commercial models trained on vast, proprietary datasets. We believe that scaling our approach with more extensive and diverse data collections could further close this gap and unlock even greater potential.

Second, our framework is designed as a flexible adaptation paradigm for existing Vision Foundation Models (VFMs). While this approach allows us to seamlessly integrate with powerful pre-trained encoders, it also means the input resolution is inherently constrained by the fixed resolution of the specific encoder chosen. Future work could focus on developing a more resolution-agnostic version of UniFlow or extending the framework to handle variable resolutions, thereby enhancing its applicability in more diverse, real-world scenarios.

# F    DECLARATION OF USE OF LARGE LANGUAGE MODELS (LLM)

We affirm that this paper was primarily written by the authors. Large Language Models (LLMs) were utilized solely as general-purpose assistive tools for language refinement, grammar correction, and stylistic improvements during the writing process. Specifically, Gemini 2.5 Flash (DeepMind, 2025) was employed for minor text polishing and rephrasing to enhance clarity and readability. No LLM was used for conceptual ideation, experimental design, data analysis, or generating any substantive content of the research.

