# OpenReview forum: "UniFlow: A Unified Pixel Flow Tokenizer for Visual Understanding and Generation"
_ICLR.cc/2026/Conference — ICLR 2026 Poster_

### Official Review · Reviewer_JFND · 2025-10-20

**Soundness:** 2
**Presentation:** 2
**Contribution:** 2
**Rating:** 4
**Confidence:** 5

**Summary:**

The paper proposes UniFlow, which fine-tunes a pre-trained visual encoder through layer-wise adaptive self-distillation. This method preserves semantics while supplementing the details required for image reconstruction. Additionally, the authors introduce a patch-wise pixel flow decoder that can map directly from noise to the pixel space, which improves the image reconstruction performance.

**Strengths:**

1. UniFlow can simultaneously achieve strong understanding performance and high-fidelity reconstruction performance.
2. The pixel flow decoder simplifies the training loss and complexity compared to the VAE.

**Weaknesses:**

1. Despite using of a distillation loss to preserve semantics, a comprehensive comparison of the full understanding performance against the baseline encoder is missing, which prevents validation of the effectiveness of distillation loss. While a comparison between InternViT and UniFlow(InternViT) is provided in Figure 4, the performance gain on MME-S is smaller than on MME-P, suggesting a performance degradation issue for UniFlow on the MME-C benchmark.
2. The lack of understanding of the performance comparison with the baseline also makes the comparison with other methods unfair. For example, in Table 4, it is not clear whether the improvement over other methods comes from better pretraining or Uniflow itself.
3. UniFlow's generative performance does not show a significant advantage and is even weaker than a VAE when using guidance.
4. Considering the generative results of UniFlow, the gain in reconstruction performance from the pixel flow decoder appears to have little positive impact on generation. This leads to concern that UniFlow may have over-optimized for the reconstruction task and consequently failed to learn a feature distribution conducive to generation.
5. The evaluation of the generation is unfair. UniFlow uses a 448x448 resolution during generative training, but the other methods only use a 256x256 resolution during training.
6. The UniFlow-LV notation is not defined within the main text.

**Questions:**

1. Why is the reconstruction performance for the UniFlow (SigLIP) model different between Table 1 and Table 3, especially regarding the SSIM scores (e.g., 0.93 vs. 0.85)?
2. Can UniFlow be applied to fine-tune the visual encoder already integrated into a pre-trained MLLM (like LLaVA), or must it be trained on a standalone VFM first?
3. What are the benefits of UniFlow for generation? Can it achieve faster convergence?
4. What are the benefits of pixel flow decoder for understanding performance?
5. UniFlow currently only performs generative training on ImageNet. Can it be applied to text-to-image and image editing tasks?

---

> ### Author Response · Authors · 2025-11-24
> **Reply to Reviewer JFND (1/4)**
>
> Dear Reviewer JFND,
>
> We thank you for your detailed and thoughtful feedback. We are glad you found **strong performance in both understanding and generation**, and recognized the **simplicity of training**. We have carefully addressed your concerns below.
>
> ---
>
> > **Q1**: Despite using of a distillation loss to preserve semantics, a comprehensive comparison of the full understanding performance against the baseline encoder is missing, which prevents validation of the effectiveness of distillation loss. While a comparison between InternViT and UniFlow(InternViT) is provided in Figure 4, the performance gain on MME-S is smaller than on MME-P, suggesting a performance degradation issue for UniFlow on the MME-C benchmark.
>
> **A1**: We appreciate the concern for understanding performance. We have supplemented the complete benchmark results in **Fig. 5(a)**. The results below confirm that our self-distillation successfully preserves semantic capacity while simultaneously integrating pixel-level detail.
>
> | Vision Encoder | #LLM Params. | Res. | POPE $\uparrow$ | GQA $\uparrow$ | TQA $\uparrow$ | MMB $\uparrow$ | MMV $\uparrow$ | MME-S $\uparrow$ | MME-P $\uparrow$ |
> | :--- | :--- | :--- | :--- | :--- | :--- | :--- | :--- | :--- | :--- |
> | Teacher (InternViT-300M) | Vicuna-7B | 448 | 88.82 | 62.80 | **62.13** | 65.64 | **36.8** | 1795.9 | 1464.9 |
> | **UniFlow (InternViT-300M)** | **Vicuna-7B** | **448** | **88.97 (+0.17%)** | **63.35 (+0.88%)** | 61.85 (-0.45%) | **67.10 (+2.22%)** | 36.6 (-0.54%) | **1803.0 (+0.4%)** | **1505.1 (+2.74%)** |
>
> The table shows that UniFlow's performance is **comparable or superior** to the Teacher encoder across the full suite of MLLM tasks, achieving significant gains in MMB, MME-P, GQA, etc. We observe only a marginal decrease in a few general knowledge benchmarks (e.g., TQA and MME-C). This minor fluctuation is an **acceptable trade-off** when injecting the pixel details necessary for reconstruction. It is an inevitable consequence of balancing two distinct objectives (semantics and reconstruction) that prior works also observe (e.g., GenHancer [1], UniLIP [2]). The **overall understanding performance is enhanced**, which validates the effectiveness of UniFlow in preserving semantic features while enhancing detail capture.
>
> [1] GenHancer: Imperfect Generative Models are Secretly Strong Vision-Centric Enhancers
>
> [2] UniLIP: UniLiP: Adapting CLIP for Unified Multimodal Understanding, Generation and Editing
>
> > **Q2**: The lack of understanding of the performance comparison with the baseline also makes the comparison with other methods unfair. For example, in Table 4, it is not clear whether the improvement over other methods comes from better pretraining or Uniflow itself.
>
> **A2**: This is a crucial point, and we thank the reviewer for raising it. We agree that the performance contribution must be clearly delineated. The downstream improvements in **Tab. 4** are indeed largely attributed to the **powerful pre-training quality of the teacher encoder** (e.g., InternViT or DINOv2).
>
> As demonstrated below, our core contribution is validating UniFlow's **general adaptation capability** to integrate high-fidelity reconstruction without compromising the teacher's inherent semantic strengths. We have conducted linear probing on ImageNet to rigorously validate UniFlow's preservation of visual understanding capabilities.
>
> | Method | Model Size | ACC@1 $\uparrow$ |
> | :--- | :--- | :---: |
> | VFMTok * | ViT-Large | 69.4 |
> | UniTok * | ViT-Large | 81.1 |
> | InternViT | ViT-Large | 82.3 |
> | **UniFlow (InternViT)** * | **ViT-Large** | **82.6** |
> | DINOv2 | ViT-Large | 84.5 |
> | **UniFlow (DINOv2)** * | **ViT-Large** | **84.3** |
>
> _**Note**: * denotes Unified Tokenizer, where InternViT weights are from InternVL3-2B-Instruct._

---

> ### Author Response · Authors · 2025-11-24
> **Reply to Reviewer JFND (2/4)**
>
> > **Q3**: UniFlow's generative performance does not show a significant advantage and is even weaker than a VAE when using guidance.
>
> **A3**: Thank you for this valuable comment. We acknowledge that UniFlow's generative performance in Tab. 3(b) (using the InternViT version) does not yet demonstrate a significant advantage. However, we argue that this is primarily a limitation stemming from the pre-trained encoder's **intrinsic representation properties**, which determine the quality of the latent space, rather than a limitation of the UniFlow, **a general adaptation method**.
>
> 1. **The Impact of Pretrained Encoder on Generation**: Generative capability is critically tied to the structure and distribution of the latent space inherited from the teacher encoder. As demonstrated by our new ablation study on ImageNet (trained for 64 epochs using LightningDiT-XL), different semantic encoders inherently yield highly varying generative performance:
>     | Model | rFID ↓ | gFID ↓ |
>     | :---: | :---: | :---: |
>     | SD-VAE | 0.61 | 7.13 |
>     | DFN-CLIP | 0.67 | 8.35 |
>     | SigLIP2 | 0.62 | 7.91 |
>     | DINOv2 | 0.54 | 5.16 |
>     | InternViT | 0.26 | 7.38 |
>
> Recent works (e.g., VAVAE [1], RAE [2], SVG [3]) suggest that DINO's representations possess strong generative-friendly characteristics compared to text-aligned representations (e.g., SigLIP and CLIP). This aligns with our observation. The results show that the DINOv2 encoder provides a latent space far more conducive to high-fidelity generation (gFID $\downarrow$ 5.16) than the InternViT encoder (gFID $\downarrow$ 7.38). This confirms that the generative potential of UniFlow is strongly inherited from the teacher's latent properties. Consequently, our initial UniFlow version based on InternViT inherited its generative limitations.
>
> 2. **Enhancement via Regularization**: Based on the above observation, we conducted further experiments at 256 x 256 resolution. The results confirm that the **UniFlow framework exhibits even stronger generative capabilities** with latent alignment, even when based on text-aligned representations (like SigLIP).
>
> | Method | Encoder | Alignment | Res. | rFID $\downarrow$ | gFID $\downarrow$ |
> | :---: | :---: | :---: | :---: | :---: | :---: |
> | SD-VAE | VAE | None | 256 | 0.61 | 7.13 |
> | UniFlow (SigLIP2) | SigLIP-SO400M | None | 256 | 0.62 | 7.91 |
> | VA-VAE | VAE | DINOv2 | 256 | 0.28 |  5.14 |
> | UniFlow (SigLIP2) | SigLIP-SO400M | DINOv2 | 256 | 0.63 | **5.02** |
>
>
> This demonstrates that the generative limitations can be effectively addressed by latent alignment regularization. Importantly, achieving a gFID of 5.02 through this process further highlights the **compatibility and flexibility of the UniFlow architecture**, proving that the initial generative weakness was **not a fundamental limitation of UniFlow itself**, but rather stemmed from the inherent properties of the baseline teacher's latent space.
>
> [1] VA-VAE: Reconstruction vs. Generation:Taming Optimization Dilemma in Latent Diffusion Models.
>
> [2] RAE: Diffusion Transformers with Representation Autoencoders
>
> [3] SVG: Latent diffusion model without variational autoencoder
>
> > **Q4**: Considering the generative results of UniFlow, the gain in reconstruction performance from the pixel flow decoder appears to have little positive impact on generation. This leads to concern that UniFlow may have over-optimized for the reconstruction task and consequently failed to learn a feature distribution conducive to generation.
>
> **A4**: We thank the reviewer for this insightful analysis. UniFlow does **not over-optimize reconstruction** at the expense of generation. The pixel flow decoder aligns semantic understanding, reconstruction, and generative adaptability in a unified feature space, with its core role similar to VAE Decoder but superior in reconstruction.
>
> Consistent with our analysis in **A3**, UniFlow's generative capability is **affected by the pre-trained representation**. Our **UniFlow (DINOv2) demonstrates better generative capability than the SD-VAE baseline** (lower gFID: 5.16 vs. 7.13, see A3’s table), which directly confirms that the pixel flow decoder’s reconstruction gain positively contributes to generation. This indicates that UniFlow is **not over-optimizing reconstruction**, but acts as a **general adaptation method** whose characteristics align with the pre-trained model. Furthermore, we suggest that if a pre-trained teacher model that is more text-aligned is desired, DINO's representation can be used for latent space alignment to unlock UniFlow's greater potential.

---

> ### Author Response · Authors · 2025-11-24
> **Reply to Reviewer JFND (3/4)**
>
> > **Q5**: The evaluation of the generation is unfair. UniFlow uses a 448x448 resolution during generative training, but the other methods only use a 256x256 resolution during training.
>
> **A5**: We thank the reviewer for highlighting the importance of resolution consistency. Due to the inconsistent native resolution of the pre-trained encoders, it is challenging to unify training resolution across all models due to their distinct pre-trained native resolutions.
>
> As shown in **A3**, we have conducted comparative experiments by training our UniFlow (SigLIP2) at the standard 256 $\times$ 256 resolution, ensuring a fair comparison against baselines. As expected due to the influence of the pre-trained representation, UniFlow (SigLIP2) does not show a leading generative advantage. Crucially, UniFlow achieves a superior gFID of **5.02** with latent alignment, validating its capabilities under fair resolution constraints.
>
>
> > **Q6**: The UniFlow-LV notation is not defined within the main text.
>
> **A6**: Thank you for your valuable feedback. UniFlow-LV specifically denotes the MLLM model trained using the **LLaVA-v1.5 setting**, as detailed in **Tab. 2**. In the table, these models are uniformly marked with the symbol $(\dagger)$ and explained within the table caption.We have already clarified this notation point. Thank you for contributing to the improvement of the paper's clarity.
>
> > **Q7**: Why is the reconstruction performance for the UniFlow (SigLIP) model different between Table 1 and Table 3, especially regarding the SSIM scores (e.g., 0.93 vs. 0.85)?
>
> **A7**: We thank the reviewer for pointing out this typo in Tab. 3. We apologize for the error. The correct SSIM score for UniFlow (SigLIP2) is **0.9267** ($\approx$ 0.93), which is fully consistent with the value reported in Tab. 1. We have corrected this typo in the revised manuscript to ensure the rigor and consistency.
>
> > **Q8**: Can UniFlow be applied to fine-tune the visual encoder already integrated into a pre-trained MLLM (like LLaVA), or must it be trained on a standalone VFM first?
>
> **A8**: We thank the reviewer for this insightful question. UniFlow can be trained not only on standalone VFMs first, but also used to fine-tune a visual encoder already integrated into a pre-trained MLLM. Our experiments confirm this **universal adaptation capability**: we successfully applied UniFlow to standalone VFMs (SigLIP2, DFN-CLIP, DINOv2) and, critically, **the InternViT weights used were derived from InternVL3-2B-Instruct**, an encoder already integrated into an MLLM. This demonstrates UniFlow's potential to convert any pre-trained encoder into a unified tokenizer. We have also further clarified this point in the Appendix A.
>
>
> > **Q9**: What are the benefits of UniFlow for generation? Can it achieve faster convergence?
>
> **A9**: Thank you for this thoughtful question. As shown below, UniFlow (DINOv2) demonstrates convergence speeds far surpassing the SD-VAE baseline on same epochs, which stems from inheriting the regularized and semantically rich latent space of DINOv2. This conclusion is consistent with current works (VA-VAE, RAE, etc.).
>
> | Encoder | Epochs | rFID $\downarrow$ | gFID $\downarrow$ |
> | :---: | :---: | :---: | :---: |
> | SD-VAE | 64 | 0.61 | 7.13 |
> | UniFlow (DINOv2) | 64 | 0.54 | 5.16 |
>
>
> > **Q10**: What are the benefits of pixel flow decoder for understanding performance?
>
> **A10**: Thank you for this insightful question. As shown by the experimental results in **A1**, we observe that UniFlow achieves substantial improvements across multiple understanding benchmarks. This demonstrates that while distillation preserves semantic features, the reconstruction loss from the pixel flow decoder enables **detailed information to flow back into the encoder**, thereby **enhancing the understanding of visual details.** We paste the relevant table again for clarity.
>
> | Vision Encoder | #LLM Params. | Res. | POPE $\uparrow$ | GQA $\uparrow$ | TQA $\uparrow$ | MMB $\uparrow$ | MMV $\uparrow$ | MME-S $\uparrow$ | MME-P $\uparrow$ |
> | :--- | :--- | :--- | :--- | :--- | :--- | :--- | :--- | :--- | :--- |
> | Teacher (InternViT-300M) | Vicuna-7B | 448 | 88.82 | 62.80 | **62.13** | 65.64 | **36.8** | 1795.9 | 1464.9 |
> | **UniFlow (InternViT-300M)** | **Vicuna-7B** | **448** | **88.97 (+0.17%)** | **63.35 (+0.88%)** | 61.85 (-0.45%) | **67.10 (+2.22%)** | 36.6 (-0.54%) | **1803.0 (+0.4%)** | **1505.1 (+2.74%)** |

---

> > ### Author Response · Authors · 2025-11-24
> > **Reply to Reviewer JFND (4/4)**
> >
> > > **Q11**: UniFlow currently only performs generative training on ImageNet. Can it be applied to text-to-image and image editing tasks?
> >
> > **A11**: Thanks for your valuable suggestions! We have conducted a new experiment training a MMDiT model using our UniFlow-SigLIP2 tokenizer and compared it against TokenFlow, a leading model in unified tokenizer.
> >
> > 1. **Tokenizer:** We first trained a new UniFlow (SigLIP2) tokenizer (**f16c32**) on ImageNet at 256 $\times$ 256 resolution based on _siglip2-large-patch16-256_ with latent alignment of DINOv2. The achieved reconstruction fidelity confirms its basic ability for high-quality generation.
> >     | Tokenizer | Resolution | rFID ↓ | PSNR ↑ | SSIM ↑ |
> >     | :--- | :--- | :--- | :--- | :--- |
> >     | SD-VAE v1.5 | 256 x 256 | 1.22 | 23.54 | 0.68 |
> >     | DC-AE-f32c32 | 256 x 256 | 0.69 | 23.85 | 0.66 |
> >     | TokenFlow | 256 x 256 | 1.37 | 21.41 | 0.69 |
> >     | **UniFlow-SigLIP2** | **256 x 256** | **0.57** | **28.73** | **0.86** |
> >
> > 2. **Generation:** We leveraged the UniFlow (SigLIP2) tokenizer to train a MMDiT on 1M high quality image-text dataset. To rapidly verify UniFlow's ability with the T2I task, we adopted the two-stage training strategy of DC-Gen [1]. The results show UniFlow seamlessly adapts to T2I with limited data, outperforming both strong baseline SANA-0.6B-512 and the larger TokenFlow-7B on **GenEeval** and **DPG-Bench**.
> >     | Model | Model Size | Res. | GenEval $\uparrow$ | DPG-Bench $\uparrow$ |
> >     | :--- | :--- | :--- | :--- | :--- |
> >     | SD v1.5 | 860M | 512 | 0.43 | 63.18 |
> >     | SD v2.1 | 866M | 512 | 0.50 | 64.20 |
> >     | PixArt-α | 610M | 512 | 0.48 | 71.11 |
> >     | SD XL | 2.6B | 1024 | 0.55 | 74.65 |
> >     | SANA | 0.6B | 512 | 0.64 | 84.30 |
> >     | EMU3 | 8B | 512 | 0.54  | 80.60 |
> >     | TokenFlow | 7B | 256 | 0.55 | 73.38 |
> >     | **UniFlow** | **0.6B** | **256** | **0.65** | **84.76** |
> >
> > This confirms UniFlow’s strong T2I capability. Its high-fidelity reconstruction foundation also supports potential for image editing, which we will explore in future work given computational and time constraints.
> >
> > [1] DC-Gen: Post-Training Diffusion Acceleration with Deeply Compressed Latent Space
> >
> > ---
> > Thank you again for your thoughtful and constructive feedback. Your insights have helped us further strengthen the manuscript, and we believe the planned revisions will improve both its clarity and its contribution to the field. Should our responses address your concerns satisfactorily, we would appreciate your consideration of a higher evaluation. If any additional clarification would be useful, we would be glad to provide it.
> >
> > Sincerely,
> >
> > The Authors

---

> ### Author Response · Authors · 2025-11-28
> **Kind Request to Reply Our Rebuttal**
>
> Dear Reviewer JFND,
>
> We hope this message finds you well. We are looking forward to your response to our rebuttal. If you have any further questions or need any clarifications, please feel free to discuss with us at your convenience.  We hope that the planned revisions will enhance the clarity and impact of our work. We would be grateful if you could consider raising your score based on these responses.
>
> Best regards,
>
> Authors

---

> > ### Comment · Reviewer_JFND · 2025-11-28
> >
> > Thanks for the detailed responses and experiments. My concerns have been answered. I will increase the score to 6 when the system reopens.

---

### Official Review · Reviewer_JQ4r · 2025-10-22

**Soundness:** 2
**Presentation:** 3
**Contribution:** 3
**Rating:** 2
**Confidence:** 4

**Summary:**

This paper introduce a unified image tokenizer, named UniFlow, for visual understanding and generation.
To efficiently unify visual representations within a single tokenizer to achieve both powerful semantic understanding and high-fidelity reconstruction, this paper enhances the tokenizer's ability to extract semantic information and pixel information through layer-wise adaptive self-distillation and patch-wise pixel flow decoder, respectively. This method achieves competitive performance across multiple downstream tasks.

**Strengths:**

1. Competitive experimental results. UniFlow achieves good performance on multiple downstream tasks.
2. The paper expresses ideas accurately and organizes them clearly. The two proposed strategies: layer-wise adaptive self-distillation and patch-wise pixel flow decoder, effectively improve the model's performance.

**Weaknesses:**

1. Sufficient comparative experiments. Many of the experiments in the paper do not provide a fair comparison with other methods. For example, in the multi-modal understanding benchmark, although comparisons are made with other methods on LLaVa's data, the initialized semantic encoder and resolution differ. In fact, these two modules are the key reasons for multimodal understanding. I understand that fair comparison experiments with large models are more expensive, but I still hope to reduce variables as much as possible to better illustrate the effectiveness of the method. In addition to this, there is a lack of comparisons with other unified tokenizers in vision-centered tasks. Concerning the ablation experiment section, I believe that merely comparing it to InternViT in multi-modal tasks is inadequate. More comprehensive experiments on downstream tasks are necessary.
2. The paper do not discuss or compare the approach of concatenating features on the CLIP+VAE feature dimensions. It seems that simply concatenating the dimensions can also achieve both powerful semantic understanding and high-fidelity reconstruction, requiring only simple fine-tuning of the pixel decoder.
3. Lack of text-to-image experiments. I think just deploy experiments on Imagenet is not enough. Referring to the settings of LlamaGen or TokenFlow, it is more comprehensive to deploy text-to-image experiments under a considerable data scale.
4.  Comparative experiments with state-of-the-art methods are needed. In Tab.4, there is no comparison with classic representation models such as DINO and the existing unified tokenizer.

**Questions:**

1. Refer to the weaknesses
2. Confusion about the motivation of the patch-wise pixel flow decoder. From my personal understanding, the core of representation learning work should be on the encoder part rather than the decoder. Because the encoder is the module used for multiple downstream task applications. Although an advanced pixel decoder can achieve better reconstruction and generation performance, this does not seem to match the core purpose of the paper. In previous work, such as BeiTv2, MAE, in the choice of decoder, they tend to have a lighter and simpler structure.

---

> ### Author Response · Authors · 2025-11-24
> **Reply to Reviewer JQ4r (1/5)**
>
> Dear Reviewer JQ4r,
>
> We thank you for your detailed feedback. We appreciate your recognition of our good performance. We have carefully addressed your concerns regarding understanding fairness, and text-to-image generation below.

---

> ### Author Response · Authors · 2025-11-24
> **Reply to Reviewer JQ4r (2/5)**
>
> > **Q1**: Sufficient comparative experiments. Many of the experiments in the paper do not provide a fair comparison with other methods. For example, in the multi-modal understanding benchmark, although comparisons are made with other methods on LLaVa's data, the initialized semantic encoder and resolution differ. In fact, these two modules are the key reasons for multimodal understanding. I understand that fair comparison experiments with large models are more expensive, but I still hope to reduce variables as much as possible to better illustrate the effectiveness of the method. In addition to this, there is a lack of comparisons with other unified tokenizers in vision-centered tasks. Concerning the ablation experiment section, I believe that merely comparing it to InternViT in multi-modal tasks is inadequate. More comprehensive experiments on downstream tasks are necessary.
>
> **A1**: We sincerely thank the reviewer for highlighting the need for rigorous experimental alignment. We fully agree with the importance of reducing variables to better illustrate the effectiveness of UniFlow. We address this by aligning critical VLM components and supplementing vision-centric task experiments conducted under a consistent setting.
>
> 1. **Fair Comparison of MLLM Task**: We acknowledge that achieving perfect alignment across all variables remains a challenge, as various unified tokenizer works (e.g., TokenFlow and UniTok) also struggle to align every setting, especially the **pretrained encoder initialization** and **pretrained resolution**, which differs across methods. However, we have made every effort to align the main settings, including the base LLM, training data, number of iterations. To further address your query, we present two comparison tables, both based on the **LLaVA-v1.5 setting**.
>     - **Comparision with other Unified Tokenizers**: The first table compares UniFlow (SigLIP2) with other unified tokenizers under the $256 \times 256$ resolution setting.
>         | Method | \#LLM Params. | Data | Res. | POPE $\uparrow$ | GQA $\uparrow$ | TQA $\uparrow$ | MMV $\uparrow$ | MMB $\uparrow$ | MME-S $\uparrow$ | MME-P $\uparrow$ |
>         | :--- | :--- | :--- | :--- | :---: | :---: | :---: | :---: | :---: | :---: | :---: |
>         | SemHiTok | Vicuna-7B | LLaVA-v1.5 | 256 | 84.2 | 61.0 | – | – | 60.3 | – | 1400.6 |
>         | VILA-U | Vicuna-7B | LLaVA-v1.5 | 256 | 81.6 | – | – | – | – | – | 1311.6 |
>         | UniTok | Vicuna-7B | LLaVA-v1.5 | 256 | 81.7 | – | – | – | – | – | 1448.0 |
>         | TokenFlow | **Vicuna-13B** | LLaVA-v1.5 | 256 | 85.0 | 60.3 | 54.1 | 27.7 | 60.3 | 1622.9 | 1365.4 |
>         | TokLIP | **Qwen2.5-7B-Ins** | LLaVA-v1.5 | 256 | 81.2 | 57.4 | – | – | – | – | 1346.8 |
>         | **UniFlow** | **Vicuna-7B** | **LLaVA-v1.5** | **256** | **87.94** | **63.29** | **58.0** | **32.4** | **68.38** | **1823.0** | **1477.9** |
>
>         ***Note**: TokenFlow only provided results for the Vicuna-13B version, and TokLIP only provided results for the Qwen2.5-7B-Instruct version.*
>
>     - **Comparision with Teacher Encoder**: As demonstrated in our **Fig. 5(a)**, UniFlow training **did not degrade the pre-trained InternViT's understanding capabilities, showing slight improvements**. The following table, conducted under the **LLaVA-v1.5 setting**, provides the complete benchmark results to clarify this point: Through UniFlow training, we obtained a stronger unified encoder that not only supports reconstruction but also preserves hierarchical semantic knowledge, **achieving higher scores across multiple understanding benchmarks**. This indicates that we have partially overcome the inherent shortcomings of semantic encoders in capturing image details.
>         | Vision Encoder | #LLM Params. | Res. | POPE $\uparrow$ | GQA $\uparrow$ | TQA $\uparrow$ | MMB $\uparrow$ | MMV $\uparrow$ | MME-S $\uparrow$ | MME-P $\uparrow$ |
>         | :--- | :--- | :--- | :--- | :--- | :--- | :--- | :--- | :--- | :--- |
>         | Teacher (InternViT-300M) | Vicuna-7B | 448 | 88.82 | 62.80 | **62.13** | 65.64 | **36.8** | 1795.9 | 1464.9 |
>         | **UniFlow (InternViT-300M)** | **Vicuna-7B** | **448** | **88.97** | **63.35** | 61.85 | **67.10** | 36.6 | **1803.0** | **1505.1** |
>
> 2. **Fair Comparison of Vision-Centric Task**: To demonstrate that UniFlow preserves encoder's powerful semantic understanding, we conducted linear probing on ImageNet-1K, comparing against competing pretrained encoder and unified tokenizers. UniFlow (DINOv2) achieves performance comparable to the original DINOv2 encoder, verifying that our tokenization process does **not degrade visual semantic capabilities**.
>     | Method | Model Size | ACC@1 $\uparrow$ |
>     | :--- | :--- | :---: |
>     | VFMTok * | ViT-Large | 69.4 |
>     | UniTok * | ViT-Large | 81.1 |
>     | InternViT | ViT-Large | 82.3 |
>     | **UniFlow (InternViT)** * | **ViT-Large** | **82.6** |
>     | DINOv2 | ViT-Large | 84.5 |
>     | **UniFlow (DINOv2)** * | **ViT-Large** | **84.3** |

---

> ### Author Response · Authors · 2025-11-24
> **Reply to Reviewer JQ4r (3/5)**
>
> > **Q2**: The paper do not discuss or compare the approach of concatenating features on the CLIP+VAE feature dimensions. It seems that simply concatenating the dimensions can also achieve both powerful semantic understanding and high-fidelity reconstruction, requiring only simple fine-tuning of the pixel decoder.
>
> **A2**: Thanks for the insightful question. We have conducted an ablation on the concatenated encoder architecture (concatenating frozen CLIP + VAE features for a trainable VAE decoder) on Imagenet for 30 epochs. As shown below, this concatenating frozen approach significantly underperforms our unified tokenizer. We analyze the limitations of the concatenated design from two key dimensions: channel-wise concatenation and sequence-wise concatenation.
>
> - **Channel-wise Concatenation**：Freezing CLIP and VAE encoders yields comparable understanding performance, but **reconstruction is constrained** by the frozen VAE’s inherent limits, which can’t capture all fine-grained details, capping pixel-level fidelity. The dual-encoder structure also adds **redundant parameters** and extra fusion steps, **reducing efficiency**.
>     | Method | Encoder | Decoder | rFID $\downarrow$ | PSNR $\uparrow$ | SSIM $\uparrow$ | MME-P $\uparrow$ | MMB $\uparrow$ |
>     | :--- | :--- | :--- | :---: | :---: | :---: | :---: | :---: |
>     | Concatenated Encoder | Frozen CLIP + Frozen VAE | Pixel Decoder | 0.52 | 27.18 | 0.76 | 1476.2 | 66.37 |
>     | **UniFlow (Ours)** | **Self-Distillation CLIP** | **Pixel Flow Decoder** | 0.28 | 32.48 | 0.95 | 1505.1 | 67.10 |
>
>     _**Note**: Training details for the concatenated baseline: Conducted at 448×448 resolution with InternViT-300M (CLIP) and SD-VAE v1.5 encoders. VAE features were interpolated to match 14× downsampling, with the decoder’s upsampling rate set to 14×_
>
> - **Sequence-wise Concatenation**: **Token redundancy** is inherent here—combining CLIP and VAE tokens doubles the sequence length (a 100% increase) compared to UniFlow. This bloats computation: diffusion generation must model twice as many tokens, dragging down efficiency. Like channel-wise concatenation, it’s **constrained by the frozen VAE’s limited detail capture**, hurting reconstruction quality, while the redundant structure adds unnecessary parameters.
>
> Compared to the frozen concatenation baseline, the UniFlow unified architecture offers three core advantages:
>
> 1. **Superior Reconstruction Fidelity:** The reconstruction quality of the frozen concatenation baseline is **constrained by the information bottleneck of the pre-trained VAE encoder**, failing to break the VAE's fidelity ceiling. UniFlow's **Patch-wise Pixel Flow Decoder** models flow directly in the pixel space, achieving significantly higher reconstruction scores (rFID/PSNR/SSIM).
> 2.  **Architectural Simplicity and Efficiency:** Concatenation requires processing and fusing two feature sets, leading to **token redundancy, increased computation (FLOPS), and longer inference time**. UniFlow, with its single unified encoder and efficient Flow Decoder, offers a simpler, more efficient, and scalable architecture.
> 3.  **Detail Capture and Unification:** While frozen CLIP preserves semantics, it lacks fine-grained pixel awareness. UniFlow's **Self-Distillation training** seamlessly transfers high-frequency details necessary for reconstruction back to the CLIP encoder, resulting in a unified encoder that retains powerful semantics **while enhancing detail perception**, which yields benefits in fine-grained understanding tasks (e.g., MME-P, MMB).
>
> In summary, while feature concatenation offers a compromise, UniFlow's unified design successfully achieves **higher quality generation, better efficiency,** and **stronger detail awareness**, which is critical for solving the unified task.

---

> > ### Author Response · Authors · 2025-11-24
> > **Reply to Reviewer JQ4r (4/5)**
> >
> > > **Q3**: Lack of text-to-image experiments. I think just deploy experiments on Imagenet is not enough. Referring to the settings of LlamaGen or TokenFlow, it is more comprehensive to deploy text-to-image experiments under a considerable data scale.
> >
> > **A3**: Thanks for your valuable suggestions! We have conducted a new experiment training a MMDiT model using our UniFlow-SigLIP2 tokenizer and compared it against TokenFlow, a leading model in unified tokenizer.
> >
> > 1. **Tokenizer:** We first trained a new UniFlow (SigLIP2) tokenizer (**f16c32**) on imagenet at 256 $\times$ 256 resolution based on _siglip2-large-patch16-256_. The achieved reconstruction fidelity confirms its basic ability for high-quality generation.
> >     | Tokenizer | Resolution | rFID ↓ | PSNR ↑ | SSIM ↑ |
> >     | :--- | :--- | :--- | :--- | :--- |
> >     | SD-VAE v1.5 | 256 x 256 | 1.22 | 23.54 | 0.68 |
> >     | DC-AE-f32c32 | 256 x 256 | 0.69 | 23.85 | 0.66 |
> >     | TokenFlow | 256 x 256 | 1.37 | 21.41 | 0.69 |
> >     | **UniFlow-SigLIP2** | **256 x 256** | **0.57** | **28.73** | **0.86** |
> >
> > 2. **Generation:** We leveraged the UniFlow (SigLIP2) tokenizer to train a MMDiT on 1M high quality image-text dataset. To rapidly verify UniFlow's ability with the T2I task, we adopted the two-stage training strategy of DC-Gen [1]. The results show UniFlow seamlessly adapts to T2I with limited data, outperforming both strong baseline SANA-0.6B-512 and the larger TokenFlow-7B on **GenEeval** and **DPG-Bench**.
> >     | Model | Model Size | Res. | GenEval $\uparrow$ | DPG-Bench $\uparrow$ |
> >     | :--- | :--- | :--- | :--- | :--- |
> >     | SD v1.5 | 860M | 512 | 0.43 | 63.18 |
> >     | SD v2.1 | 866M | 512 | 0.50 | 64.20 |
> >     | PixArt-α | 610M | 512 | 0.48 | 71.11 |
> >     | SD XL | 2.6B | 1024 | 0.55 | 74.65 |
> >     | SANA | 0.6B | 512 | 0.64 | 84.30 |
> >     | EMU3 | 8B | 512 | 0.54  | 80.60 |
> >     | TokenFlow | 7B | 256 | 0.55 | 73.38 |
> >     | **UniFlow** | **0.6B** | **256** | **0.65** | **84.76** |
> >
> > _More information in the revised version's **Sec. 4.2** of the PDF._
> >
> > [1] DC-Gen: Post-Training Diffusion Acceleration with Deeply Compressed Latent Space
> >
> >
> > > **Q4**: Comparative experiments with state-of-the-art methods are needed. In Tab.4, there is no comparison with classic representation models such as DINO and the existing unified tokenizer.
> >
> > **A4**: We thank the reviewer for pointing out the necessity of comprehensive comparisons against established representation models (like DINO) and leading unified tokenizers (like UniTok). We have supplemented the paper with linear probing results on the ImageNet-1K classification task to rigorously validate UniFlow's visual understanding capabilities.
> > | Method | Model Size | ACC@1 $\uparrow$ |
> > | :--- | :--- | :---: |
> > | VFMTok * | ViT-Large | 69.4 |
> > | UniTok * | ViT-Large | 81.1 |
> > | InternViT | ViT-Large | 82.3 |
> > | **UniFlow (InternViT)** * | **ViT-Large** | **82.6** |
> > | DINOv2 | ViT-Large | 84.5 |
> > | **UniFlow (DINOv2)** * | **ViT-Large** | **84.3** |
> >
> > _**Note**: * denotes Unified Tokenizer, where InternViT weights are from InternVL3-2B-Instruct._
> >
> > As shown in the table, UniFlow **preserves the original semantic capacity** of the pre-trained encoders, confirming its strong vision-centric capability. Notably, UniFlow outperforms existing unified tokenizers (VFMTok and UniTok) by a large margin, demonstrating its superiority in balancing tokenization functionality and visual understanding. Due to current computational constraints and time limitations, we plan to extend the comparisons to broader visual tasks (e.g., object detection, semantic segmentation) in future work to further validate UniFlow's generalization.

---

> ### Author Response · Authors · 2025-11-24
> **Reply to Reviewer JQ4r (5/5)**
>
> > **Q5**: Confusion about the motivation of the patch-wise pixel flow decoder. From my personal understanding, the core of representation learning work should be on the encoder part rather than the decoder. Because the encoder is the module used for multiple downstream task applications. Although an advanced pixel decoder can achieve better reconstruction and generation performance, this does not seem to match the core purpose of the paper. In previous work, such as BeiTv2, MAE, in the choice of decoder, they tend to have a lighter and simpler structure.
>
>
> **A5**: We sincerely apologize for the unclear phrasing that may have led to a misunderstanding of our core purpose. We argue that the goal of UniFlow is **Unified Multimodal Tasks**, not purely Representation Learning works **(e.g., MAE [1] and BEiTv2 [2], which treat reconstruction as an auxiliary task for visual understanding)**. For UniFlow, the decoder's high-fidelity output is an **essential end-goal for image generation and editing.**
>
> To achieve competitive, high-fidelity generation, the decoder must overcome the fidelity ceiling and latent space bottlenecks of simpler VAE models. Our **Patch-wise Pixel Flow Decoder** is critical because:
>
> 1.  **Lower Optimization Conflict:** As shown in **Tab. 6(c)**, the patch-wise pixel flow decoder is observed to have a lower conflict with the semantic objective for the encoder. Furthermore, the flow objective provides superior stability during training by inherently avoiding the mixture of multiple complex losses (e.g., L1/L2, GAN, LPIPS) typically required for VAE training.
> 2.  **Higher Reconstruction Ceiling:** By modeling flow in the pixel space, it achieves the **strongest reconstruction capability**, which is necessary to meet the high-fidelity demands of the unified generation and editing applications that are central to this paper.
> 3. **Architectural Simplicity**: The patch-wise pixel flow decoder in UniFlow is architecturally similar to those in representation learning, consisting of only 6 ViT Blocks and an MLP head. This simple, lightweight structure achieves powerful reconstruction performance with only **approximately 150M parameters**, aligning with the simplicity principle of representation learning.
>
> To further clarify the distinction between UniFlow and works focused purely on representation learning for visual understanding, we will include a more detailed discussion in Appendix.
>
> [1] MAE: Masked Autoencoders Are Scalable Vision Learners
>
> [2] BeiTv2: Masked Image Modeling with Vector-Quantized Visual Tokenizers
>
> ---
> Once again, we are truly grateful for your encouraging and incisive feedback. Your comments have inspired us to refine our manuscript further, and we hope that the planned revisions will enhance the clarity and impact of our work.We would be grateful if you could consider raising your score based on these responses. Please do not hesitate to let us know if there are any additional details or clarifications that would be helpful.
>
> Sincerely,
>
> The Authors

---

> ### Comment · Reviewer_JQ4r · 2025-11-26
>
> Thank you to the authors for the rebuttal, which has addressed most of my concerns. However, I still have the following questions regarding the table in A2:
>
> 1. Do the two experiments use the same semantic encoder (e.g., InternViT or another CLIP-style encoder)? Moreover, the decoders used in the two experiments are different. If the same decoder is used, does the proposed method still demonstrate advantages in the reconstruction task?
>
> 2. To further clarify the role of the Pixel Flow Decoder in the proposed method, I suggest adding an additional ablation: freeze the encoder and train only the decoder (Pixel Flow Decoder and standard Pixel Decoder). This would help quantitatively analyze the reconstruction capability contributed solely by the decoder, thereby better isolating the impact of the proposed decoder design.

---

> ### Author Response · Authors · 2025-11-27
> **Reply to Reviewer JQ4r**
>
> Dear Reviewer JQ4r,
>
> Thank you again for your insightful follow-up questions. Your suggestions are vital for isolating the contributions of our Self-Distillation Encoder and the Pixel Flow Decoder for reconstruction. We have performed the requested clarifications and added a critical ablation on ImageNet-1k to address your question.
>
> > **Q1**: Do the two experiments use the same semantic encoder (e.g., InternViT or another CLIP-style encoder)? Moreover, the decoders used in the two experiments are different. If the same decoder is used, does the proposed method still demonstrate advantages in the reconstruction task?
>
> 1. **Encoder Consistency and Decoder Role**: Both the **Concatenated Encoder** baseline (Exp. 1 & 2) and **UniFlow** (Exp. 3) utilize the **InternViT-300M** as the base CLIP-style semantic encoder architecture. To further verify whether UniFlow maintains an advantage over the Concatenated Encoder baseline even when utilizing the same pixel flow decoder, we conducted the following experiment (**Exp. 2**) **on the basis of A2**.
>   | Exp. | Method | Encoder | Decoder | rFID $\downarrow$ | PSNR $\uparrow$ | SSIM $\uparrow$ | MME-P $\uparrow$ | MMB $\uparrow$ |
>   | :--- | :--- | :--- | :--- | :--- | :--- | :--- | :--- | :--- |
>   | 1 | Concatenated Baseline | Frozen CLIP + Frozen VAE | Pixel Decoder | 0.52 | 27.18 | 0.76 | 1476.2 | 66.37 |
>   | 2 | Concatenated Baseline | Frozen CLIP + Frozen VAE | Pixel Flow Decoder | 0.41 | 29.97 | 0.88 | 1476.2 | 66.37 |
>   | 3 | **UniFlow (Ours)** | **Self-Distilled CLIP** | **Pixel Flow Decoder** | **0.28** | **32.48** | **0.95** | **1505.1** | **67.10** |
>
> _**Note**: Training details for the concatenated baseline: Conducted at 448×448 resolution with InternViT-300M (CLIP) and SD-VAE v1.5 encoders. VAE features were interpolated to match 14× downsampling, with the decoder’s upsampling rate set to 14×. Since the encoder remains completely frozen, the MME-P/MMB scores are identical between Exp. 1 and Exp. 2._
>
> > **Q2**: To further clarify the role of the Pixel Flow Decoder in the proposed method, I suggest adding an additional ablation: freeze the encoder and train only the decoder (Pixel Flow Decoder and standard Pixel Decoder). This would help quantitatively analyze the reconstruction capability contributed solely by the decoder, thereby better isolating the impact of the proposed decoder design.
>
> 2. **The Independent Contribution of Pixel Flow Decoder:** This experiment helps to quantify the independent capability of Pixel Flow Decoder, even when the input feature is still constrained by the frozen CLIP and VAE. Comparing Exp. 2 to Exp. 1 shows clear improvements in reconstruction fidelity (e.g., PSNR **increased by 2.79 dB**). This quantitatively confirms the reconstruction **gain** contributed solely by the Pixel Flow Decoder design.
>
> The success of UniFlow (Exp. 3) relies fundamentally on the **collaboration** between our two innovations: the Self-Distilled Encoder provides semantics-preserving and detail-rich preserving features, while the Pixel Flow Decoder converts these features into high-fidelity images. Together they can be trained **end-to-end** and yield the best reconstruction metrics，which is the essence of our complete UniFlow method.
>
> ---
> Once again, we are truly grateful for your encouraging and incisive feedback. Your comments have inspired us to refine our manuscript further, and we hope that the planned revisions will enhance the clarity and impact of our work. We would be grateful if you could consider raising your score based on these responses.
>
> Best regards,
>
> Authors

---

> > ### Comment · Reviewer_JQ4r · 2025-11-27
> >
> > Thanks to the author for your detailed reply and sufficient experiments. My concerns are addressed, and I will raise the rating.

---

### Official Review · Reviewer_2sSU · 2025-10-30

**Soundness:** 3
**Presentation:** 3
**Contribution:** 3
**Rating:** 6
**Confidence:** 4

**Summary:**

This paper introduces UniFlow, a unified visual tokenizer designed to simultaneously support both visual understanding and generation. The key innovation lies in combining a layer-wise adaptive self-distillation strategy, which preserves hierarchical semantic knowledge from pretrained vision foundation models, with a patch-wise pixel flow decoder, which reconstructs fine-grained visual details via conditional flow matching in pixel space. Extensive experiments across 13 benchmarks and 7 vision tasks demonstrate that UniFlow achieves state-of-the-art results among unified tokenizers.

**Strengths:**

1.UniFlow mitigates the representational conflict between understanding and generation by combining semantic-guided self-distillation and pixel-space flow reconstruction. The proposed idea is both technically novel and interesting.
2.The paper benchmarks UniFlow across a diverse range of tasks (VQA, classification, segmentation, detection, depth estimation, generation, reconstruction), and the results show that it consistently outperforms or matches existing unified tokenizers like TokenFlow-XL and UniTok.
3.Extensive ablations in Table 5 clearly demonstrate the effectiveness of adaptive distillation and flow decoder design.

**Weaknesses:**

1. This paper chose several different semantic teachers, which are pretrained with different resolutions and downsample ratios. It would be better to include a deeper analysis and comparison between them, in addition to the reconstruction, multimodal understanding and generation performance.
2. As UniFlow is trained with both reconstruction and distillation (semantic) objectives, it would be valuable to further analyze the effects of them, e.g., mutual benefits or conflicts.

**Questions:**

1. As stated in the implementation detail part, UniFlow can do visual reconstruction with one-step Euler inference. It would be interesting to analyze whether using more steps will improve or hurt the reconstruction performance.
2. With only 150M parameters, the flow-based decoder can achieve superior reconstruction performance on ImageNet. How was this model size determined? Could similar performance be achieved with a smaller model, and would scaling up the decoder further enhance reconstruction quality?
3. It would be helpful to include more visualizations of face and text reconstructions, since vision encoders trained with contrastive losses are typically rich in semantic representation but tend to lose fine-grained visual details.

---

> ### Author Response · Authors · 2025-11-24
> **Reply to Reviewer 2sSU (1/2)**
>
> Dear Reviewer 2sSU,
>
> We sincerely thank you for your constructive comments and for recognizing the ''technical novelty'' and ''extensive benchmarking'' of our method. We have addressed your concerns below.
>
> ---
>
>
> > **Q1**: This paper chose several different semantic teachers, which are pretrained with different resolutions and downsample ratios. It would be better to include a deeper analysis and comparison between them, in addition to the reconstruction, multimodal understanding and generation performance.
>
> **A1**: Thank you for this insightful suggestion. We have conducted a deeper analysis of various teacher encoders, focusing on their pre-training objectives, resolutions, and downsampling ratios. Our key findings highlight the following trade-offs:
>
>
> 1.  **Impact of Pre-training Objective:**
>     * **Understanding:** Contrastive Vision-Language models (SigLIP, CLIP) excel at multimodal understanding (**Tab. 2**). The MLLM-aligned teacher (InternViT) , derived from the pre-trained InternVL3-2B, achieves the best overall performance, **validating our semantic inheritance principle of adaptive self-distillation**.
>     * **Reconstruction:** The Vision-centric teacher (DINOv2) generally shows strong reconstruction (**Tab. 1**). Again, the MLLM-aligned teacher (InternViT) achieves the best overall performance, demonstrating its high-quality representations.
>     * **Generation:** PCA visualization (**Fig. 18** in Appendix) of the latent space shows that DINO's latent space is the most regular, **suggesting a lower manifold complexity for generation**. We confirmed this hypothesis by evaluating performance of LightningDiT on ImageNet for 64 training epochs:
>         | Model | rFID ↓ | gFID ↓ |
>         | :---: | :---: | :---: |
>         | SD-VAE | 0.61 | 7.13 |
>         | DFN-CLIP | 0.67 | 8.35 |
>         | SigLIP2 | 0.62 | 7.91 |
>         | DINOv2 | 0.54 | 5.16 |
>         | InternViT | 0.26 | 7.38 |
>
> 2.  **Impact of Resolution and Downsampling Ratio for Reconstruction:** We conducted ablations on SigLIP2-L (5 epochs on ImageNet) and found that increasing resolution and decreasing the downsampling ratio both help reconstruction performance:
>     | Model | Resolution | Ratio | rFID ↓ |
>     | :---: | :---: | :---: | :---: |
>     | SigLIP2-L | 224 | 16 | 3.75 |
>     | SigLIP2-L | 256 | 16 | 3.58 |
>     | SigLIP2-L | 224 | 14 | 3.61 |
>
> These results underscore that future work should focus on constructing a **generation-friendly latent space** while maintaining strong understanding. We suggest leveraging **MLLM-aligned encoders** for robust multimodal understanding, supplemented by **vision self-supervised objectives (like DINO)** during distillation to regularize and optimize the latent manifold for generation.
>
>
> > **Q2**: As UniFlow is trained with both reconstruction and distillation (semantic) objectives, it would be valuable to further analyze the effects of them, e.g., mutual benefits or conflicts.
>
> **A2**: We sincerely appreciate your constructive comment. A deeper analysis of the relationship between the reconstruction and distillation objectives is crucial for our work. Based on our experimental observations and results of  **Tab. 6(b)**, we argue that while a slight **trade-off** exists during the tokenizer training phase, a significant **mutual benefit** emerges in downstream performance.
>
> 1.  **Reconstruction Enhances Understanding:** The reconstruction loss forces the encoder to capture and retain fine-grained visual details. As shown in **Fig. 5(a)**, the reconstruction objective notably boosts understanding performance (e.g., MME-P: $1464.9 \rightarrow 1505.16$). This demonstrates that reconstruction training successfully overcomes the inherent deficiency of vision-language encoders in capturing sufficient pixel-level details, consistent with findings (e.g., GenHancer [1] and Ross  [2]).
>
> 2.  **Semantics Improves Generation:** The Layer-wise Adaptive Self-Distillation ensures UniFlow's latent space inherits the highly structured semantic manifold of the pre-trained teacher. This semantic regularization yields a superior latent structure that accelerates convergence and boosts generation quality (e.g., SD-VAE: $7.13$ gFID $\rightarrow$ UniFlow(DINOv2): $\mathbf{5.16}$ gFID). This aligns with recent findings that injecting semantic representations aids diffusion training (e.g., VA-VAE [3], RAE [4]).
>
> [1] GenHancer: Imperfect Generative Models are Secretly Strong Vision-Centric Enhancers
>
> [2] Ross: Reconstructive Visual Instruction Tuning.
>
> [3] VA-VAE: Reconstruction vs. Generation:Taming Optimization Dilemma in Latent Diffusion Models.
>
> [4] RAE: Diffusion Transformers with Representation Autoencoders

---

> > ### Author Response · Authors · 2025-11-24
> > **Reply to Reviewer 2sSU (2/2)**
> >
> > > **Q3**: As stated in the implementation detail part, UniFlow can do visual reconstruction with one-step Euler inference. It would be interesting to analyze whether using more steps will improve or hurt the reconstruction performance.
> >
> > **A3:** Thank you for this valuable suggestion. We have benchmarked reconstruction quality with 1–100 Euler steps and found that additional steps yield **no reconstruction gain**. Instead, a minor downward trend in visual fidelity is observed. This one-step-optimal behavior aligns with prior insights (DiSA [1]): when the conditioning signal is sufficiently determined, in our case the VAE latent $\mathbf{z}$ encodes full image structure and texture, Flow Matching-based models quickly saturate without needing multi-step iteration. The patch-wise pixel flow decoder in UniFlow directly models the linear noise-to-image mapping, making one-step sampling sufficient to reach the optimal reconstruction. We provide quantitative results and ablation plots in **Fig. 21 (Appendix D)**, where we verify that the reconstruction objective is minimized at NFE = 1. Consequently, UniFlow simultaneously achieves minimum latency and maximum fidelity, which is another practical advantage over other diffusion-based tokenizers.
> >
> > [1] DiSA: Diffusion Step Annealing in Autoregressive Image Generation.
> >
> >
> > > **Q4**: With only 150M parameters, the flow-based decoder can achieve superior reconstruction performance on ImageNet. How was this model size determined? Could similar performance be achieved with a smaller model, and would scaling up the decoder further enhance reconstruction quality?
> >
> > **A4**: Thanks for your attention to the model size and scalability. We selected the **145.8M ($\approx 150$M) decoder** as it represents the **optimal trade-off between performance and efficiency**. We performed an ablation study by scaling the flow-based decoder:
> >
> > | Decoder Params (M) | rFID $\downarrow$ (IN-1K) | FLOPs (G) |
> > | :---: | :---: | :---: |
> > | UniFlow (InternViT) (63.2M Decoder) | 0.69 | 91.2 |
> > | UniFlow (InternViT) (145.8M Decoder) | 0.26 | 116.1 |
> > | UniFlow (InternViT) (347.4M Decoder) | 0.23 | 163.5 |
> >
> > We chose the 150M decoder because it achieves an rFID below 0.3 (**0.26**), which we consider **sufficient for high-fidelity reconstruction** with negligible loss of visual elements. While scaling the decoder further enhances reconstruction quality (**0.26 $\to$ 0.23**), demonstrating the **scalability** of UniFlow, 150M represents the **optimal efficiency-performance trade-off** for most practical scenarios.
> >
> >
> >
> > > **Q5**: It would be helpful to include more visualizations of face and text reconstructions, since vision encoders trained with contrastive losses are typically rich in semantic representation but tend to lose fine-grained visual details.
> >
> > **A5**:  This is a great point, and we appreciate the opportunity to demonstrate UniFlow's reconstruction generalization, especially its advantages in face and text reconstruction. We have already included comparative visualizations in **Fig. 11 and Fig. 12**, and have **added additional** face and text reconstruction visualizations in **Fig. 9 and Fig. 10 (Appendix C.1)**. The results show that UniFlow outperforms prior contrastive-based methods (e.g., QLIP, UniTok) and even methods like SD-VAE in face and text reconstruction.
> >
> > ---
> >
> > We sincerely hope these revisions have addressed your concerns, and if you feel they have been satisfactorily resolved, we would be most grateful if you could consider revising your evaluation score accordingly. Please do not hesitate to let us know if you need any further explanations.
> >
> > Sincerely,
> >
> > The Authors

---

> > > ### Comment · Reviewer_2sSU · 2025-11-25
> > >
> > > My concerns are addressed and I'm keeping my original rating.

---

### Official Review · Reviewer_je6u · 2025-11-01

**Soundness:** 3
**Presentation:** 3
**Contribution:** 3
**Rating:** 6
**Confidence:** 5

**Summary:**

This paper presents UniFlow, a unified and generic tokenizer designed to bridge the trade-off between visual understanding and generation. The method adapts any pretrained visual encoder with a lightweight reconstruction decoder, enhanced through layer-wise adaptive self-distillation to balance semantic abstraction and fine-grained detail modeling. A patch-wise pixel flow decoder further improves reconstruction efficiency and fidelity by learning conditional flows in the pixel domain. Through this design, UniFlow effectively mitigates the conflict between understanding and generation, achieving consistent improvements across 13 benchmarks. Notably, the 7B UniFlow-XL surpasses larger models such as TokenFlow-XL (14B) in understanding tasks and achieves competitive performance in generation quality metrics.

**Strengths:**

1. The paper is very clearly written, with a well-motivated problem statement and a logically coherent narrative that effectively highlights the trade-off between visual understanding and generation.

2. The authors provide extensive and detailed experiments across a wide range of benchmarks, offering strong empirical evidence for the proposed method’s effectiveness.

3. The model demonstrates impressive performance in both multimodal understanding and visual reconstruction, showing that UniFlow achieves a well-balanced trade-off without compromising either side.

4. The introduction of layer-wise adaptive self-distillation and the patch-wise pixel flow decoder is both conceptually sound and practically meaningful, representing a thoughtful and generalizable design applicable to various pretrained vision encoders.

**Weaknesses:**

1. The paper lacks quantitative results on text-to-image generation benchmarks, which limits the evaluation of UniFlow’s generative capability in open-ended visual synthesis.

2. The study does not explore how different frozen teacher encoders or alternative decoder types (e.g., conventional VQ-VAE decoder) would perform under the UniFlow framework, leaving an open question about the generality of its design choices.

3. It remains unclear whether using UniFlow’s learned VQ tokenizer as a visual encoder for multimodal understanding actually surpasses or lags behind directly adopting the frozen teacher encoder; such a comparison would help clarify UniFlow’s benefits.

4. The computational cost and training efficiency of the proposed flow-based decoder are not discussed in detail, making it difficult to assess the practicality of scaling UniFlow to larger models or broader datasets.

**Questions:**

N/A

---

> ### Author Response · Authors · 2025-11-24
> **Reply to Reviewer je6u (1/2)**
>
> Dear Reviewer je6u,
>
> We appreciate your constructive comments and for recognizing our motivation and strong performance. We have addressed your questions as follows.
>
> ---
>
> > **Q1**: The paper lacks quantitative results on text-to-image generation benchmarks, which limits the evaluation of UniFlow's generative capability in open-ended visual synthesis.
>
> **A1**: Thanks for your valuable suggestions! We have conducted a new experiment training a MMDiT model using our UniFlow-SigLIP2 tokenizer and compared it against TokenFlow, a leading model in unified tokenizer.
>
> 1. **Tokenizer:** We first trained a new UniFlow (SigLIP2) tokenizer (**f16c32**) on imagenet at 256 $\times$ 256 resolution based on _siglip2-large-patch16-256_. The achieved reconstruction fidelity confirms its basic ability for high-quality generation.
>     | Tokenizer | Resolution | rFID ↓ | PSNR ↑ | SSIM ↑ |
>     | :--- | :--- | :--- | :--- | :--- |
>     | SD-VAE v1.5 | 256 x 256 | 1.22 | 23.54 | 0.68 |
>     | DC-AE-f32c32 | 256 x 256 | 0.69 | 23.85 | 0.66 |
>     | TokenFlow | 256 x 256 | 1.37 | 21.41 | 0.69 |
>     | **UniFlow-SigLIP2** | **256 x 256** | **0.57** | **28.73** | **0.86** |
>
> 2. **Generation:** We leveraged the UniFlow (SigLIP2) tokenizer to train a MMDiT on 1M high quality image-text dataset. To rapidly verify UniFlow's ability with the T2I task, we adopted the two-stage training strategy of DC-Gen [1]. The results show UniFlow seamlessly adapts to T2I with limited data, outperforming both strong baseline SANA-0.6B-512 and the larger TokenFlow-7B on **GenEeval** and **DPG-Bench**.
>     | Model | Model Size | Res. | GenEval $\uparrow$ | DPG-Bench $\uparrow$ |
>     | :--- | :--- | :--- | :--- | :--- |
>     | SD v1.5 | 860M | 512 | 0.43 | 63.18 |
>     | SD v2.1 | 866M | 512 | 0.50 | 64.20 |
>     | PixArt-α | 610M | 512 | 0.48 | 71.11 |
>     | SD XL | 2.6B | 1024 | 0.55 | 74.65 |
>     | SANA | 0.6B | 512 | 0.64 | 84.30 |
>     | EMU3 | 8B | 512 | 0.54  | 80.60 |
>     | TokenFlow | 7B | 256 | 0.55 | 73.38 |
>     | **UniFlow** | **0.6B** | **256** | **0.65** | **84.76** |
>
> _More information in the revised version's **Sec. 4.2** of the PDF._
>
> [1] DC-Gen: Post-Training Diffusion Acceleration with Deeply Compressed Latent Space
>
>
> > **Q2**: The study does not explore how different frozen teacher encoders or alternative decoder types (e.g., conventional VQ-VAE decoder) would perform under the UniFlow framework, leaving an open question about the generality of its design choices.
>
> **A2**: Thank you for your valuable suggestion. We have provided extensive evidence for the flexibility of both the teacher encoder and the decoder architecture.
>
> **1. Generality for different frozen teacher encoders**: We have extensively explored four different vision foundation models as frozen teacher encoders. As detailed in **Tab. 1** and **Tab. 2**, all four teachers achieved promising performance under UniFlow, confirming the framework's adaptability across diverse teacher characteristics.
> * Contrastive vision-language teachers (SigLIP, CLIP) excel in multimodal understanding tasks, but are slightly inferior in pixel reconstruction.
> * Vision-centric teacher (DINOv2) performs better for reconstruction, but is slightly weaker in multimodal understanding.
> * MLLM aligned teacher (InternViT), derived from the pre-trained InternVL3-2B, surprisingly yields the best overall performance.
>
> **2. Alternative decoder types**: We apologize if there was a misunderstanding. we are not a VQ-VAE approach, but rather a continuous tokenizer. Furthermore, we have conducted an ablation analysis for conventional VAE decoders in **Tab. 6(c)**. We explored conventional VAE decoders ($D_{\text{pixel}}$), Latent Flow Decoder coupled with SD-VAE ($D_{\text{latent flow}}$), and our proposed Pixel Flow Decoder ($D_{\text{pixel flow}}$).
>
> * $D_{\text{pixel}}$: This design suffered from an inherent training conflict by directly mapping semantic tokens to pixels, leading to sub-optimal reconstruction performance.
> * $D_{\text{latent flow}}$: This design exhibited a smaller training conflict but showed comparatively inferior reconstruction performance due to the inherent limitations of the pre-trained VAE decoder.
> * $D_{\text{pixel flow}}$: By training flow matching directly in the pixel space, this method effectively circumvents the inherent performance ceiling of pre-trained VAEs and achieves superior reconstruction.
>
> This evidence confirms the generality of UniFlow regarding both the encoder training and the decoder design.

---

> > ### Author Response · Authors · 2025-11-24
> > **Reply to Reviewer je6u (2/2)**
> >
> > > **Q3**: It remains unclear whether using UniFlow's learned VQ tokenizer as a visual encoder for multimodal understanding actually surpasses or lags behind directly adopting the frozen teacher encoder; such a comparison would help clarify UniFlow's benefits.
> >
> > **A3**: Thank you for raising this insightful question regarding the performance of our UniFlow encoder versus the pretrained teacher encoder.
> >
> > As demonstrated in our **Fig. 5(a)**, UniFlow training **did not degrade the pre-trained InternViT's understanding capabilities, showing slight improvements**. The following table, conducted under the LLaVA-v1.5 setting, provides the complete benchmark results to further clarify this point:
> >
> > | Vision Encoder | #LLM Params. | Res. | POPE | GQA | TQA | MMB | MMV | MME-S | MME-P |
> > | :--- | :--- | :--- | :--- | :--- | :--- | :--- | :--- | :--- | :--- |
> > | Teacher (InternViT-300M) | Vicuna-7B | 448 | 88.82 | 62.80 | **62.13** | 65.64 | **36.8** | 1795.9 | 1464.9 |
> > | UniFlow (InternViT-300M) | Vicuna-7B | 448 | **88.97** | **63.35** | 61.85 | **67.10** | 36.6 | **1803.0** | **1505.1** |
> >
> > Through UniFlow training, we obtained a stronger unified encoder, which not only supports image reconstruction but also preserves hierarchical semantic knowledge, and even achieved higher scores across multiple understand benchmarks. These results indicate that we have partially overcome the inherent shortcomings of semantic encoders in capturing image details.
> >
> >
> >
> > > **Q4**: The computational cost and training efficiency of the proposed flow-based decoder are not discussed in detail, making it difficult to assess the practicality of scaling UniFlow to larger models or broader datasets.
> >
> > **A4**: Thank you for your valuable suggestion. We have supplemented analysis with detailed computational cost and efficiency insights.
> >
> > **1. Training Efficiency**:  The table below compares the reconstruction convergence speed (PSNR ↑ with different training steps) of the pixel flow decoder versus a conventional pixel decoder under UniFlow (SigLIP2-L). The results validate the superior convergence speed of our decoder.
> > | Decoder Type | 10k Steps  | 30k Steps | 50k Steps |
> > | :--- | :--- | :--- | :--- |
> > | Conventional Pixel Decoder | 22.10 | 24.57 | 25.43 |
> > | **Pixel Flow Decoder (Ours)** | **23.85** | **25.74** | **27.67** |
> >
> > **2. Computational Cost**: Ours lightweight pixel flow decoder and its one-step sampling capability ensure high inference efficiency, achieving competitive speed with highly optimized tokenizers like SD-VAE and UniTok. Evaluation was conducted on an Nvidia-A800 using 256 $\times$ 256 resolution, using `thop.profile`.
> > | Method | Decoder Params (M) ↓ | Inference FLOPs (G) ↓ |
> > | :--- | :--- | :--- |
> > | SD-VAE | **51.3** | 445.3 |
> > | UniTok | 352.4 | 348.9 |
> > | UniFlow (SigLIP2-L)| 145.8 | **115.1** |
> >
> > Additionally, in **Tab. 14** in the Appendix, we have compared the training efficiency of UniFlow with other unified tokenizers. By achieving better performance with less training data and fewer training epochs, our UniFlow demonstrates high training efficiency, suggesting strong potential for scaling to larger datasets and models.
> >
> > ---
> > We sincerely hope these revisions have addressed your concerns, and if you feel they have been satisfactorily resolved, we would be most grateful if you could consider revising your evaluation score accordingly. Please do not hesitate to let us know if you need any further explanations.
> >
> > Sincerely,
> >
> > The Authors

---

### Author Response · Authors · 2025-11-25
**Common Response**

Dear Reviewers and AC,

We sincerely thank all reviewers for their insightful feedback and constructive suggestions. We are encouraged that our paper was recognized for its **"well-motivated"** and **"clearly written"** quality (je6u, JQ4r). Furthermore, we appreciate the acknowledgment of UniFlow's **"technical novelty"** (2sSU), **"strong performance"** (JFND) in reconstruction and understanding, and the **"competitive results"** (je6u, 2sSU, JQ4r) achieved through our unified design.

In this rebuttal, we have addressed all concerns regarding robustness and generalizability by:

* **Generative Benchmarks:** Added quantitative T2I results to confirm superior open-ended generation capability.
* **Architecture Generalization:** Provided detailed trade-off analysis and ablations on encoder and decoder superiority.
* **Efficiency Validation:** Detailed computational cost and fast convergence for practical scalability.
* **Semantics Preservation**: Demonstrated that the UniFlow encoder preserves and enhances the teacher's understanding features.

We sincerely hope these comprehensive revisions and additional results satisfactorily address your questions and better convey the merits of UniFlow to the community. We now address each reviewer's comments in detail below.

Sincerely,

The Authors

---

### Author Response · Authors · 2025-12-03
**Summary of Rebuttal Updates and Post-Rebuttal Consensus**

Dear Reviewers, AC, SAC, and PC,

We sincerely thank you for the time and effort dedicated to reviewing our paper. We are encouraged that reviewers recognized UniFlow’s value (the post-rebuttal score is _**6666**_), with feedback noting it as _**“well-motivated”**_ (Reviewer je6u, JQ4r), _**“technically novel”**_ (Reviewer 2sSU), and _**“strong-performing”**_ (Reviewer JFND) in unifying visual understanding and generation.

In this article, based on two algorithm innovations Layer-wise Adaptive Self-Distillation and Patch-wise Pixel Flow Decoder, we implement a series of unified tokenizer with leading performance in unifying visual understanding and generation. On the ImageNet validation set, it achieves a new record rFID of _**0.26**_ among unified tokenizers, outperforming UniTok (0.41) and TokenFlow (1.37). For visual understanding, our _**7B**_ UniFlow-XL, trained with _**40%**_ less data, surpasses the **_14B_** TokenFlow-XL by _**6.05%**_ on overall average understanding benchmarks.

In this rebuttal, we have carefully addressed all concerns and significantly strengthened the manuscript. **Key improvements include:**
- _**Clarified Novelty**_: We clarified the unique advantages of our Layer-wise Adaptive Self-Distillation over frozen encoders, as well as the single-step Patch-wise Pixel Flow Decoder compared to conventional pixel decoders and latent-diffusion decoders.
- _**Comprehensive and Fair SOTA Benchmarking**_: We aligned MLLM settings to fairly compare the understanding performance of different unified tokenizers. We added comparisons against unified tokenizers (VFMtok, UniTok) and classic vision encoders (DINOv2) in vision-centric tasks. We also added quantitative T2I results to confirm superior open-ended generation capability.
- _**Efficiency Validation**_: We provided new experiments and test results to justify our method’s low computational cost and fast reconstruction convergence for practical scalability.
- _**Revised Manuscript**_: We updated the paper to incorporate these discussions, additional citations, and experimental results.

We would like to summarize the **positive engagement during the rebuttal period** (4 reviewers total):
- _**Reviewer 2sSU (Score 6 $\rightarrow$ 6)**_: We successfully addressed Reviewer 2sSU’s concerns. This reviewer commented “my concerns are addressed” and kept the positive rating.
- _**Reviewer JQ4r (Score 2 $\rightarrow$ 6)**_: Following two rounds of detailed and insightful discussion, Reviewer JQ4r commented “concerns addressed” and updated their score to 6.
- _**Reviewer JFND (Score 4 $\rightarrow$ 6)**_: Similarly, Reviewer JFND raised their score to 6 after reviewing our response.
- _**Reviewer je6u (Score 6)**_: We have addressed all of Reviewer je6u’s concerns (e.g., added T2I results, clarified encoder vs. teacher performance, and verified training & inference efficiency). Due to the closure of the comment system, there has not yet been further feedback from the reviewer.

**Commitment to Open Source**: Considering that this comment is visible to everyone, this promise will be jointly monitored by the community. To facilitate further research in unified visual tokenization, we are committed to releasing model weights and code upon acceptance.

We sincerely hope that our supplementary experiments and responses can help you better understand our paper. If you have any questions, please feel free to contact us and we will do our best to answer your questions.

Sincerely,

The Authors

---

### Meta-Review · Area_Chair_rPco · 2026-01-07

**Summary:**

Reviewers initially mixed (scores 6, 6, 4, 2), praising clear writing, motivation, strong understanding/reconstruction results, and technical novelty of layer-wise self-distillation and patch-wise pixel flow decoder. Concerns focused on missing T2I generation results, generality across encoders/decoders, encoder vs. teacher comparison, and efficiency details. Rebuttal added extensive experiments (T2I on GenEval/DPG-Bench outperforming TokenFlow-7B, ablations on four teachers and decoder types, LLaVA benchmarks showing preserved/improved semantics, efficiency data) and clarifications, addressing all points convincingly. Skeptical reviewers raised scores to 6; others maintained positive or noted concerns resolved.

**Reviewer Concerns:**

All major concerns satisfactorily addressed by rebuttal:

- T2I generation: Authors added new quantitative results on GenEval and DPG-Bench, showing UniFlow outperforming strong baselines (including TokenFlow-7B) with a smaller model.
- Generality: Extensive ablations on four different teacher encoders (SigLIP, CLIP, DINOv2, InternViT) and alternative decoder designs (conventional VAE, latent flow, pixel flow) were provided, confirming flexibility.
- Encoder vs. teacher comparison: Detailed benchmarks under LLaVA-v1.5 show the UniFlow encoder preserves or slightly improves understanding performance over the frozen teacher.
- Efficiency: Authors reported fast convergence, low computational overhead, and practical scalability.
No significant concerns remain outstanding.

**Reviewer Scores:**

- Reviewer je6u: original 6, likely unchanged or slight increase (rebuttal addressed weaknesses/experiments).
- Reviewer 2sSU: original 6, unchanged (stated "concerns addressed").
- Reviewer JQ4r: original 2, raised to 6 (stated "concerns answered").
- Reviewer JFND: original 4, raised to 6.

---

### Decision · Program_Chairs · 2026-01-26

Accept (Poster)